# GeoGCD: Geometry-Guided Hierarchical Learning for Generalized Category Discovery

## Abstract

Generalized Category Discovery (GCD) requires learning a feature space in which both known and previously unseen categories are well organised across multiple semantic granularities. Existing hierarchical approaches rely either on fixed taxonomic priors, which are blind to the actual geometry of the learned features, or on data-driven prototypes, which discard the prior semantic structure; cross-level alignment is typically enforced through KL-based losses that become uninformative or unstable precisely when novel-class predictions are most fragile. We argue that the global geometry of semantics, the shape that taxonomic relations carve in feature and label space, is the natural object to be preserved in this setting, and we introduce a **Geo**metry-Guided Hierarchical Learning framework for **G**eneralized **C**ategory **D**iscovery (**GeoGCD**) that operationalises this principle. GeoGCD models pairwise relations through a hybrid soft-label matrix that fuses taxonomic similarity with a diffusion similarity derived from random walks on the batch graph, and aligns predictions across granularities via a Wasserstein consistency loss for which we establish formal guarantees of finiteness, continuity, and gradient informativeness under disjoint support. On standard fine-grained GCD benchmarks, GeoGCD achieves new state-of-the-art accuracy on CUB-200-2011 and Stanford Cars with **79.93** and **80.10** in overall, respectively, while notably enhancing performance on known classes through better preservation of global semantic geometry with **87.73** and **93.57** in Old accuracy. Code is available at: `https://anonymous.4open.science/r/GeoGCD-7C66`.

## 1 Introduction

Generalized Category Discovery (GCD) is fundamentally a representation learning problem: given partially labelled data that contains both known and previously unseen classes, the goal is to learn a feature space in which categories at multiple semantic granularities are simultaneously well separated and well organised, even when no supervision is available for the novel ones (Vaze et al., 2022a; Wen et al., 2023; He et al., 2025). A representation that supports category discovery in fine-grained domains must therefore satisfy two distinct requirements at once. First, it must respect semantic structure: samples that belong to the same family or order should sit in coherent regions of the feature space, reflecting the prior knowledge encoded in naturally occurring taxonomies (He et al., 2025). Second, it must preserve geometric coherence: the global arrangement of clusters, including those formed by novel classes, must reflect the actual density and connectivity of the data manifold, not only the local angular relations between features (Nadler et al., 2005; Rastegar et al., 2023; Wang et al., 2023).

Existing hierarchical GCD methods consistently address one side at the expense of the other. A first family relies on fixed taxonomic priors (Wang et al., 2023; He et al., 2025) and weights pairwise relations through tree-distance similarity, which gives a clean prior signal but is, by construction, blind to the geometry of the learned features: any cluster structure that the encoder discovers beyond the taxonomy, particularly for novel classes, is invisible to the supervision signal. A second family infers hierarchies directly from data (Rastegar et al., 2023; Hao et al., 2023) or organises representations through prototype-level contrast (Wen et al., 2023; Wang et al., 2024b), which captures local geometry but discards the explicit semantic prior, leaving the

representation free to drift towards arrangements that are inconsistent with the underlying taxonomy. Cross-granularity consistency, when present at all, is typically enforced through KL-based losses that align coarse predictions with those propagated from finer levels. This choice is comfortable in the standard supervised setting, but it is fragile in category discovery: when novel-class predictions are noisy and the predicted and pseudo-coarse distributions concentrate on disjoint regions of the label simplex, KL is either infinite or carries gradients that ignore the spatial separation of supports (Arjovsky et al., 2017), and the cross-level signal silently degrades exactly when it should do the most work. The gap, in our view, is conceptual rather than algorithmic: the global geometry of semantics, the shape that taxonomic relations carve in feature and label space, is rarely treated as a first-class object of learning (He et al., 2025; Wang et al., 2023; Rastegar et al., 2023; Wen et al., 2023; Nadler et al., 2005).

We take this geometry as the guiding principle for the representation. Concretely, we model semantic similarity at every granularity as the fusion of two complementary geometries. A taxonomic similarity, derived from shortest-path distances on the hierarchy tree, encodes prior knowledge that is stable but blind to data. A diffusion similarity, defined through $t$-step random walks on the batch graph of normalised features and a Gaussian kernel, encodes the data-driven manifold geometry that aggregates information across all paths of length $t$, robust to noise and to elongated cluster shapes, and intrinsically multi-scale via the spectrum (Nadler et al., 2005). The same geometric principle extends naturally to alignment across levels. Since the predicted coarse distribution and the pseudo-coarse distribution induced by the fine-level prediction can have nearly disjoint supports during training, particularly over novel classes, we replace the standard KL term in the cross-granularity consistency by the 1-Wasserstein distance. The distance is finite, continuous, and gradient-rich on the entire label simplex and explicitly encodes the geometric separation between distributions. Fundamentally, KL divergence ignores the underlying metric structure of the label space, whereas the Wasserstein distance explicitly respects it.

We propose GeoGCD (Hierarchical Geometric Generalized Category Discovery), a diffusion-guided semantic contrastive framework that turns the global geometry of semantics into both a representation-learning target and a cross-level alignment criterion. Our contributions are threefold:

- A new formulation of hierarchical category discovery in which the global geometry of semantics, beyond fixed taxonomic priors and local pairwise similarity, drives representation learning.

- A hybrid soft-label matrix combining taxonomic and diffusion-based similarities for hierarchical contrastive learning, together with a 1-Wasserstein cross-granularity consistency loss with formal guarantees of finiteness, continuity, and gradient informativeness under disjoint support.

- Consistent gains on standard fine-grained GCD benchmarks, supported by ablations that isolate the role of each geometric component.

## 2 Related Work

**Generalized Category Discovery.** Novel Category Discovery (NCD) (Han et al., 2019) introduced the transfer clustering paradigm, in which knowledge from labelled known-class data is used to cluster unlabelled images coming from entirely unseen categories (Han et al., 2020; 2021; Jia et al., 2021; Zhong et al., 2021). Generalized Category Discovery (GCD) (Vaze et al., 2022a) relaxes the closed-world assumption and allows the unlabelled pool to contain samples from both known and novel classes, which makes the problem closer to a real open-world setting. A number of follow-up works have pushed GCD along different directions (Wang et al., 2023; Hao et al., 2023; Pu et al., 2023; Joseph et al., 2022; Cendra et al., 2024; Wang et al., 2024a). SimGCD (Wen et al., 2023) proposes a simple end-to-end parametric framework that combines contrastive learning with entropy-regularised classification, so it no longer needs a post-hoc clustering step. SPTNet (Wang et al., 2024b) adds spatial prompt tuning to make the backbone focus more on discriminative regions. DebGCD (Liu & Han, 2025) addresses label bias and semantic shift between known and novel classes through an OOD-based debiased classifier together with a certainty-driven curriculum. PromptCAL (Zhang et al., 2023) calibrates contrastive affinities with the help of auxiliary prompts. Despite this progress, most of these methods still operate at a single semantic granularity: each image is assigned to exactly one class, and the multi-level taxonomic structure that naturally exists in many fine-grained domains is largely ignored.

**Hierarchical Learning for Category Discovery.** A few works (Wei et al., 2021; Chang et al., 2021; Du et al., 2021; Qu et al., 2016) have started to bring hierarchical structure into GCD in order to go beyond the single-granularity setting. InfoSieve (Rastegar et al., 2023) and CiPR (Hao et al., 2023) infer abstract hierarchies directly from the data via iterative semi-supervised clustering, so they build a coarse-to-fine tree without using any external taxonomy. TIDA (Wang et al., 2023) extends the open-world semi-supervised setting by defining prototypes at several manually chosen granularity levels and forcing the predictions across levels to stay consistent. More recently, SEAL (He et al., 2025) relies on naturally occurring biological taxonomies, order, family, and species, to guide a multi-task contrastive framework across granularities. SEAL has two main components: a Hierarchical Semantic-Guided Soft Contrastive loss, which weights negative pairs by their taxonomic distance instead of treating all negatives the same way, and a Cross-Granularity Consistency (CGC) module, which minimises the KL divergence between a fine-level prediction and a pseudo-coarse distribution obtained by passing the fine-level softmax through a learnable species-to-family mapping matrix. GeoGCD builds on this hierarchical formulation but departs by replacing fixed similarity and probability-based alignment with geometry-aware similarity and metric-based consistency.

**Diffusion Maps and Manifold-Based Similarity.** Diffusion maps (Nadler et al., 2005) give a principled way to study the intrinsic geometry of high-dimensional data through random walks on a weighted graph and are applied in machine learning for representation learning (Farghly et al., 2025; Wang & Pehlevan, 2025; Argov & Wagner, 2025). The diffusion distance is defined as the $\ell_2$ distance between rows of the $t$-step Markov transition matrix, so it accumulates information across all paths of length $t$ between two points. This makes the resulting metric robust to small perturbations, naturally multi-scale, and more sensitive to global cluster topology than to single pairwise distances. GeoGCD uses diffusion-based geometry as a supervisory signal that directly guides hierarchical contrastive learning, allowing the representation to reflect emerging density modes of novel classes.

**Optimal Transport and Wasserstein Distance.** The Wasserstein distance (Arjovsky et al., 2017) measures the minimum cost of transporting probability mass from one distribution to another under a ground metric on the sample space. Compared to KL divergence, Wasserstein distance stays finite and informative even when the two distributions have completely disjoint support, and the geometry it induces respects the underlying metric structure. This property has made it popular in generative modelling (Arjovsky et al., 2017), domain adaptation, and knowledge distillation. GeoGCD leverages this property for cross-granularity alignment, where predicted and pseudo-coarse distributions frequently have disjoint support during novel-class discovery.

## 3 Preliminary

### 3.1 Generalized Category Discovery

We follow the standard GCD setting introduced in (Vaze et al., 2022a). Let $\mathcal{D}_l = \{(x_i, y_i)\}_{i=1}^{N_l}$ denote the labelled set, where each image $x_i \in \mathcal{X}$ has a class label $y_i \in \mathcal{Y}_l$, and let $\mathcal{D}_u = \{x_j\}_{j=1}^{N_u}$ denote the unlabelled set, whose underlying labels lie in $\mathcal{Y}_u \supseteq \mathcal{Y}_l$. The novel-class set is $\mathcal{Y}_n = \mathcal{Y}_u \setminus \mathcal{Y}_l$. The total number of classes $|\mathcal{Y}_u| = K$ is either given or estimated by an off-the-shelf method (Vaze et al., 2022a).

The goal is to learn an encoder $f_\theta : \mathcal{X} \to \mathbb{R}^d$ together with a classification head $g_\phi : \mathbb{R}^d \to \Delta^{K-1}$ that produces correct predictions on both known and novel classes:

$$\min_{\theta, \phi} \ \mathbb{E}_{(x,y) \sim \mathcal{D}_l}[\mathcal{L}_{\text{sup}}(g_\phi(f_\theta(x)), y)] \ + \ \lambda \, \mathbb{E}_{x \sim \mathcal{D}_l \cup \mathcal{D}_u}[\mathcal{L}_{\text{unsup}}(f_\theta(x))], \tag{1}$$

where $\mathcal{L}_{\text{sup}}$ is a supervised loss on the labelled set, $\mathcal{L}_{\text{unsup}}$ is an unsupervised representation loss applied on all images, and $\lambda > 0$ balances the two terms. At evaluation time, accuracy is reported on three splits of the unlabelled set: ALL, OLD (samples whose label is in $\mathcal{Y}_l$), and NEW (samples whose label is in $\mathcal{Y}_n$).

**Hierarchical extension.** When a semantic taxonomy is available, each class $y$ at the finest granularity is also associated with labels at $H-1$ coarser levels. We write $y^{(h)} \in \mathcal{Y}^{(h)}$ for the label at level $h$, with

$h = H$ being the finest level and $h = 1$ the coarsest, and we denote the full hierarchical label as $y^{(1:H)} = (y^{(1)}, \ldots, y^{(H)})$. Hierarchical GCD aims to predict $y^{(1:H)}$ for every image, and to keep predictions across levels consistent, e.g. a sample predicted as a given species should also be predicted as the corresponding family.

## 3.2 Revisiting the Baseline: Soft Contrastive Learning and Cross-Granularity Alignment

We briefly revisit two design conventions, used by SEAL (He et al., 2025) among others, that our method takes as a starting point.

**Soft contrastive learning.** Standard contrastive losses use a hard $\{0, 1\}$ assignment between positive and negative pairs. A common refinement is to soften this assignment through a similarity matrix $\mathbf{Y}^{(h)} \in [0, 1]^{B \times B}$ derived from the taxonomy, where the shortest-path distance between two labels is converted into a graded similarity. This matrix replaces the binary mask in an InfoNCE-style loss and serves as the soft supervision signal at granularity $h$.

**Cross-granularity alignment.** To keep predictions consistent across levels, a learnable mapping matrix $\mathbf{M}^{(h)} \in [0, 1]^{K_H \times K_h}$ from fine to coarse classes (rows summing to one) maps the fine-level prediction $p_i^{(H)}$ to a pseudo-coarse distribution $\tilde{p}_i^{(h)} = (\mathbf{M}^{(h)})^\top p_i^{(H)}$, which is then aligned with the directly predicted $p_i^{(h)}$ by minimising the KL divergence $\mathrm{KL}(p_i^{(h)} \| \tilde{p}_i^{(h)})$.

Both conventions are taken as a baseline; our contributions lie in how we construct $\mathbf{Y}^{(h)}$ (Section 4.2) and the divergence we use to align $p_i^{(h)}$ with $\tilde{p}_i^{(h)}$ (Section 4.3).

## 3.3 Diffusion Maps

Diffusion maps (Nadler et al., 2005) are a non-linear dimensionality reduction and similarity-analysis tool based on the spectrum of a Markov chain defined on the data graph.

**Construction.** Let $\mathbf{Z} = [z_1, \ldots, z_N]^\top \in \mathbb{R}^{N \times d}$ be a batch of $\ell_2$-normalised features. We first build a Gaussian affinity matrix $\mathbf{W} \in \mathbb{R}^{N \times N}$,

$$W_{ij} = \exp\left(-\frac{\|z_i - z_j\|_2^2}{\sigma^2}\right), \tag{2}$$

where $\sigma > 0$ is a bandwidth (a common choice is the median pairwise distance). We then row-normalise $\mathbf{W}$ to obtain a Markov transition matrix $\mathbf{P} = \mathbf{D}^{-1}\mathbf{W}$, where $\mathbf{D} = \mathrm{diag}\left(\sum_j W_{ij}\right)$, so that $P_{ij}$ is the one-step probability of jumping from $i$ to $j$.

**Diffusion distance.** For an integer scale $t \geq 1$, the $t$-step transition matrix $\mathbf{P}^t$ has entries $[\mathbf{P}^t]_{ij}$ equal to the probability of going from $i$ to $j$ in exactly $t$ steps. The *diffusion distance* between two points $i$ and $j$ at scale $t$ is the weighted $\ell_2$ distance between the corresponding rows of $\mathbf{P}^t$:

$$D_t^2(i, j) = \sum_{k=1}^{N} \frac{\left([\mathbf{P}^t]_{ik} - [\mathbf{P}^t]_{jk}\right)^2}{\pi_k}, \tag{3}$$

where $\boldsymbol{\pi}$ is the stationary distribution of $\mathbf{P}$. The diffusion distance integrates information over all paths of length $t$ joining $i$ and $j$, which makes it less sensitive to noise and to elongated cluster shapes than the raw cosine distance. Two points are close under $D_t$ when many short walks of length $t$ connect them, which captures membership in the same density mode rather than mere geometric proximity.

## 3.4 1-Wasserstein Distance

We finally recall the definition of the Wasserstein distance and the closed form that will be used in the rest of the paper.

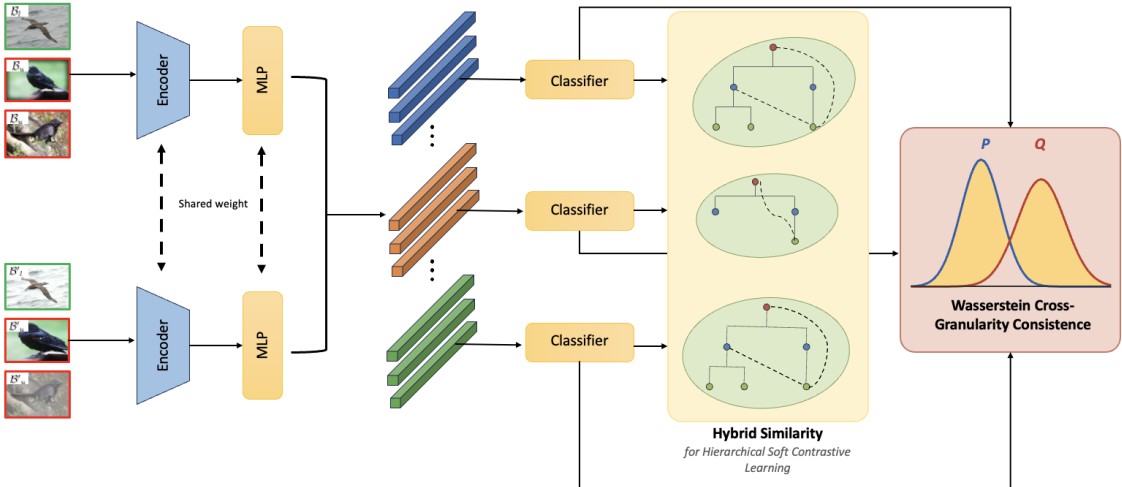

Figure 1: **Overview of the proposed GeoGCD framework.** Two augmented views of each batch are encoded by a shared encoder–MLP backbone and split into per-level feature stacks (one per granularity). Each level feeds a classifier and the **Hybrid Similarity** module, which fuses taxonomic and diffusion-based signals into a soft label matrix for hierarchical contrastive learning. The classifier predictions are then aligned by the **Wasserstein Cross-Granularity Consistency** module.

**General definition (Villani, 2008; Peyré & Cuturi, 2019)** Given two probability measures $p$ and $q$ on a metric space $(\mathcal{Z}, c)$ with ground cost $c : \mathcal{Z} \times \mathcal{Z} \to \mathbb{R}_{\geq 0}$, the 1-Wasserstein distance is

$$W_1(p, q) = \inf_{\gamma \in \Pi(p,q)} \int_{\mathcal{Z} \times \mathcal{Z}} c(z, z') \, \mathrm{d}\gamma(z, z'), \tag{4}$$

where $\Pi(p, q)$ is the set of joint distributions (couplings) with marginals $p$ and $q$.

**Closed form on ordinal label spaces (Bonneel et al., 2014; Vallender, 1974).** When the support is one-dimensional and ordinal, e.g. $\mathcal{Z} = \{1, 2, \ldots, K\}$ with the natural metric $c(k, k') = |k - k'|$, $W_1$ admits a simple closed form in terms of the cumulative distribution functions:

$$W_1(p, q) = \sum_{k=1}^{K-1} \big| F_p(k) - F_q(k) \big|, \qquad F_p(k) = \sum_{j \leq k} p(j). \tag{5}$$

This expression is fully differentiable in both $p$ and $q$ and can be evaluated in $\mathcal{O}(K)$ time, which makes it convenient as a training loss.

## 4 Method

### 4.1 GeoGCD: Geometry-Guided Hierarchical Generalized Category Discovery

Building on the advantages of semantic hierarchies, we propose the **Hi**erarchical **Geo**metric framework for **GCD** (**GeoGCD**). The overall framework is outlined in Fig. 1. Unlike prior GCD methods that rely on single-granularity supervision (Vaze et al., 2022a; Wen et al., 2023; Wang et al., 2024b; Liu & Han, 2025), infer abstract hierarchies from data (Rastegar et al., 2023; Wang et al., 2023), or guide learning with a fixed taxonomy tree under KL-based alignment (He et al., 2025), GeoGCD grounds hierarchical learning in the geometry of two complementary spaces, the feature manifold and the label simplex, through three components: (1) a multi-task backbone producing level-specific features and classifiers; (2) a hybrid soft contrastive loss that fuses taxonomic similarity with a manifold-aware diffusion similarity (Nadler et al.,

2005), capturing both prior structure and the evolving cluster geometry of novel classes; and (3) a Wasserstein cross-granularity consistency objective that remains finite and gradient-rich even when the predicted and inferred coarse distributions have nearly disjoint support, the typical regime for novel classes early in training. The remainder of this section details the two new modules in turn: the hybrid contrastive supervision (Section 4.2) and the Wasserstein cross-granularity consistency loss (Section 4.3).

### 4.2 Hybrid Similarity for Hierarchical Soft Contrastive Learning

For each input $x_i$, the encoder $f_\theta$ produces a backbone feature $\mathbf{e}_i$, which is routed through $H$ projection heads to give level-specific features $\mathbf{f}_i^{(h)} = \pi^{(h)}(\mathbf{e}_i)$, $h = 1, \ldots, H$, all $\ell_2$-normalised. At each granularity, we want a soft label matrix that drives contrastive learning by reflecting how related two samples are. A purely taxonomic signal carries strong prior structure but ignores the current geometry of the embedding; a purely manifold-based signal does the opposite. We build this module to leverage the utilities of them.

**Taxonomic similarity.** Inside a mini-batch of size $B$, we build a per-level taxonomic similarity $\mathbf{Y}_{\text{hier}}^{(h)} \in \mathbb{R}^{B \times B}$ as a weighted combination of cosine similarities at every granularity, giving more weight to the target level and progressively less to coarser ones. Concretely, with $\mathbf{S}_{ij}^{(h)} = (\mathbf{f}_i^{(h)})^\top \mathbf{f}_j^{(h)}$, we set

$$\mathbf{Y}_{\text{hier}}^{(H)} = \alpha \mathbf{S}^{(H)} + \beta \mathbf{S}^{(H-1)} + \gamma \mathbf{S}^{(1)}, \tag{6}$$

and analogous mixtures for the coarser levels, with $\alpha \geq \beta \geq \gamma \geq 0$. Each $\mathbf{Y}_{\text{hier}}^{(h)}$ is then min-max normalised and made row-stochastic.

**Diffusion similarity.** The taxonomic matrix is mostly a function of the prior structure and is blind to the actual cluster geometry of the learned features. We complement it with a diffusion-based similarity that aggregates information along random walks on the batch graph. At each level $h$, we form a Gaussian affinity

$$\mathbf{W}_{ij}^{(h)} = \exp\left(-\|\mathbf{f}_i^{(h)} - \mathbf{f}_j^{(h)}\|_2^2 / \sigma_W^2\right), \tag{7}$$

row-normalise it into a Markov transition matrix $\mathbf{P}^{(h)} = (\mathbf{D}^{(h)})^{-1}\mathbf{W}^{(h)}$, and apply $t$ diffusion steps. The diffusion distance between samples $i$ and $j$ at scale $t$ is the $\ell_2$ distance between the corresponding rows of $(\mathbf{P}^{(h)})^t$, and converting it to a similarity through a Gaussian kernel and row-normalising yields

$$[\mathbf{Y}_{\text{diff}}^{(h)}]_{ij} \propto \exp\left(-\left\|[(\mathbf{P}^{(h)})^t]_{i,:} - [(\mathbf{P}^{(h)})^t]_{j,:}\right\|_2 / \sigma_D\right). \tag{8}$$

Two samples receive high diffusion similarity when many short walks of length $t$ on the batch graph connect them, that is, when they belong to the same density mode. This captures the structure of *novel* classes as soon as they form coherent clusters, even before their taxonomic position is known.

**Hybrid similarity and contrastive loss.** We fuse the two through a convex combination with blend coefficient $\lambda \in [0, 1]$ and re-normalise:

$$\mathbf{Y}_{\text{hybrid}}^{(h)} = \text{rownorm}\left(\lambda \mathbf{Y}_{\text{hier}}^{(h)} + (1 - \lambda) \mathbf{Y}_{\text{diff}}^{(h)}\right). \tag{9}$$

The two limits recover known designs: $\lambda = 1$ gives a purely taxonomic signal, and $\lambda = 0$ relies entirely on the data manifold. The hybrid matrix replaces the hard $\{0, 1\}$ assignment in a standard InfoNCE-style loss; with temperature $\tau$, the soft contrastive loss at level $h$ is

$$\mathcal{L}_{\text{con}}^{(h)} = -\frac{1}{B} \sum_{i=1}^{B} \sum_{j \neq i} [\mathbf{Y}_{\text{hybrid}}^{(h)}]_{ij} \log \frac{\exp\left((\mathbf{f}_i^{(h)})^\top \mathbf{f}_j^{(h)} / \tau\right)}{\sum_{k \neq i} \exp\left((\mathbf{f}_i^{(h)})^\top \mathbf{f}_k^{(h)} / \tau\right)}. \tag{10}$$

Pairs with high hybrid similarity contribute strongly to the gradient, while pairs with low similarity behave as hard negatives, giving a graded supervision signal that adapts both to the taxonomy and to the evolving feature geometry. The diffusion matrix is detached from the computation graph; gradients flow only through the features $\mathbf{f}_i^{(h)}$ inside the contrastive term. Small values of $t$ ($t \in \{2, 3\}$) are sufficient in practice: larger $t$ collapses the random walk towards the stationary distribution and erases the cluster structure we want to preserve.

### 4.3 Wasserstein Cross-Granularity Consistency

The second module ties together predictions made at different granularities. We adopt the mapping-matrix design reviewed in Section 3.2: at each coarse level $h$, the model produces a directly predicted coarse distribution $p_i^{(h)} = g_\phi^{(h)}(\mathbf{e}_i) \in \Delta^{K_h-1}$ and a pseudo-coarse distribution $\tilde{p}_i^{(h)} = (\mathbf{M}^{(h)})^\top p_i^{(H)}$ obtained by routing the fine-level prediction through the learnable mapping matrix $\mathbf{M}^{(h)}$. The matrix $\mathbf{M}^{(h)}$ is initialised from the taxonomy (a one-hot row for each known fine class, a uniform row for each unseen fine class) and refined during training by an exponential moving average of the model's coarse predictions on confident samples, so that the fine-to-coarse routing of novel classes is learned rather than fixed. Cross-granularity consistency requires the two distributions to agree, since they describe the same underlying coarse class of $x_i$. Where our design departs from prior work is in the choice of divergence used to enforce this agreement.

**Wasserstein objective.** Building on the properties reviewed in Section 3.4, we replace the standard KL term with the 1-Wasserstein distance, which on the ordinal coarse-class space with ground metric $c(k, k') = |k - k'|$ admits the closed form

$$\mathcal{L}_{\mathrm{W-CGC}}^{(h)} = W_1\big(p_i^{(h)}, \tilde{p}_i^{(h)}\big) = \sum_{k=1}^{K_h-1} \big|F_{p_i^{(h)}}(k) - F_{\tilde{p}_i^{(h)}}(k)\big|, \tag{11}$$

with $F_p(k) = \sum_{j \le k} p(j)$. This expression is differentiable in both arguments, runs in $\mathcal{O}(K_h)$ time, The cumulative form introduces an ordering over the coarse indices. This ordering merely reflects the taxonomy's arbitrary index order rather than an intrinsic ranking of the classes. A column-permutation test and a substitution with permutation-invariant divergences in Appendix C give consistent results on CUB and Stanford Cars, showing that our results do not depend on the coarse ordering. The qualitative claim that $W_1$ stays bounded, continuous and gradient-rich under disjoint support is formalised in Lemma 1 and Propositions 1 and 2 in the appendix.

### 4.4 Final Training Objective

The full training objective combines a per-level supervised cross-entropy on the labelled set, the hybrid soft contrastive loss in equation 10, and the Wasserstein cross-granularity consistency loss in equation 11:

$$\mathcal{L} = \sum_{h=1}^{H} \Big(\mathcal{L}_{\mathrm{sup}}^{(h)} + \mu\,\mathcal{L}_{\mathrm{con}}^{(h)}\Big) + \nu \sum_{h=1}^{H-1} \mathcal{L}_{\mathrm{W-CGC}}^{(h)}, \tag{12}$$

with scalar weights $\mu, \nu > 0$.

Algorithm 1 summarises a single training epoch of GeoGCD, making explicit both how the hybrid similarity enters the contrastive loss and how the mapping matrices are updated via EMA.

### 4.5 Formal Analysis of the Wasserstein Cross-Granularity Objective

This section collects the formal statements supporting the discussion in Section 4.3; the proofs are deferred to Appendix C.4. Throughout, we work on a finite ordinal label space $\mathcal{Y} = \{1, \ldots, K\}$ with ground metric $c(k, k') = |k - k'|$, and write $F_p(k) = \sum_{j \le k} p(j)$ for the cumulative distribution function of $p \in \Delta^{K-1}$.

Second, the statements below describe the disjoint-support regime that is typical at initialisation. They are meant as motivation for choosing $W_1$ over KL, not as a description of the full training dynamics. We extend this to the training dynamics of the consistency term in Appendix C.5 (Proposition 3).

**Lemma 1** (Disjoint support is the typical regime at initialisation). *Consider a coarse level $h$ with $K_h$ classes. Assume that at initialisation $p_i^{(h)}$ is approximately concentrated on a single coarse index drawn uniformly from $\{1, \ldots, K_h\}$, and that the pseudo-coarse distribution $\tilde{p}_i^{(h)}$, obtained by routing the fine-level prediction through a randomly initialised mapping matrix $\mathbf{M}^{(h)}$, is approximately concentrated on a coarse index drawn independently and uniformly from the same set. Then*

$$\Pr\big[\mathrm{supp}(p_i^{(h)}) \cap \mathrm{supp}(\tilde{p}_i^{(h)}) = \emptyset\big] \ge 1 - \frac{1}{K_h}. \tag{13}$$

---

**Algorithm 1** GeoGCD training epoch

---

**Require:** encoder $f_\theta$, per-level heads $g_\phi^{(h)}$, mapping matrices $\{\mathbf{M}^{(h)}\}$, taxonomy tree, weights $\mu, \nu$, blend $\lambda$, diffusion steps $t$, current epoch $e$

1: **for** each mini-batch $\{x_i\}$ **do**
2:      $\mathbf{e}_i \leftarrow f_\theta(x_i)$;   split into per-level features $\mathbf{f}_i^{(h)}$
3:      $\mathbf{Y}_{\text{hier}} \leftarrow$ taxonomic similarity;   $\mathbf{Y}_{\text{diff}} \leftarrow t$-step random walk on the batch graph
4:      $\mathbf{Y} \leftarrow \lambda \, \mathbf{Y}_{\text{hier}} + (1-\lambda) \, \mathbf{Y}_{\text{diff}}$                            ▷ hybrid soft labels
5:      $\mathcal{L}_{\text{con}}^{(h)} \leftarrow$ soft contrastive loss with $\mathbf{Y}$;   $\mathcal{L}_{\text{sup}}^{(h)} \leftarrow$ CE on labelled samples
6:      $p_i^{(h)} \leftarrow \text{softmax}(g_\phi^{(h)}(\mathbf{e}_i))$;   $\tilde{p}_i^{(h)} \leftarrow (\mathbf{M}^{(h)})^\top p_i^{(H)}$               ▷ pseudo-coarse
7:      $\mathcal{L}_{\text{W-CGC}}^{(h)} \leftarrow W_1\big(p_i^{(h)}, \tilde{p}_i^{(h)}\big)$
8:      $\mathcal{L} \leftarrow \sum_h (\mathcal{L}_{\text{sup}}^{(h)} + \mu \, \mathcal{L}_{\text{con}}^{(h)}) + \nu \sum_{h<H} \mathcal{L}_{\text{W-CGC}}^{(h)}$
9:      update $\theta, \phi$ by gradient descent on $\mathcal{L}$
10: **end for**
11: **if** $e \geq$ warm-up **then**
12:      refine each $\mathbf{M}^{(h)}$ by EMA of confident coarse predictions of novel fine classes
13: **end if**
14: **return** $f_\theta, \{g_\phi^{(h)}\}, \{\mathbf{M}^{(h)}\}$

---

*More generally, if both distributions concentrate on $s$ coarse indices each, chosen uniformly at random and independently, the probability of disjoint supports is at least $1 - s^2/K_h$.*

**Proposition 1** (Boundedness and continuity under disjoint support)**.** *Let $p, q \in \Delta^{K-1}$.*

1. *$W_1(p, q) \leq K - 1$, and $W_1$ is continuous on $\Delta^{K-1} \times \Delta^{K-1}$.*

2. *There exist sequences $p_n, q_n$ with disjoint supports such that $\text{KL}(p_n \,\|\, q_n) = +\infty$ for all $n$, while $W_1(p_n, q_n) \to 0$.*

**Proposition 2** (Non-vanishing gradient at distributional separation)**.** *Let $p_\theta \in \Delta^{K-1}$ be smoothly parametrised by $\theta \in \mathbb{R}^d$ and let $q \in \Delta^{K-1}$ be fixed. Suppose $\text{supp}(p_{\theta_0}) \cap \text{supp}(q) = \emptyset$.*

1. *$\text{KL}(q \,\|\, p_{\theta_0}) = +\infty$ (its gradient is undefined), and $\text{KL}(p_{\theta_0} \,\|\, q)$ is finite only if $p_{\theta_0}$ vanishes wherever $q$ does, a condition that fails under disjoint support. When the latter is finite, its gradient depends only on the overlapping support and carries no information about the spatial separation between $\text{supp}(p_{\theta_0})$ and $\text{supp}(q)$.*

2. *The partial gradient of $W_1$ with respect to $p(j)$ has the explicit form*

$$\frac{\partial W_1(p, q)}{\partial p(j)} \;=\; \sum_{k=j}^{K-1} \text{sign}\big(F_p(k) - F_q(k)\big), \tag{14}$$

*which is non-zero whenever $p \neq q$, and whose magnitude grows with the number of indices at which the CDFs disagree, i.e. with the spatial separation between $\text{supp}(p)$ and $\text{supp}(q)$.*

## 5 Experiments

### 5.1 Experimental Setup

**Datasets.** We conduct experiments on three fine-grained image classification benchmarks that admit a natural multi-level taxonomy, following the Semantic Shift Benchmark (SSB) protocol (Vaze et al., 2022b) **CUB-200-2011** (Wah et al., 2011) (200 bird species), **Stanford Cars** (Krause et al., 2013) (196 car models), and **FGVC-Aircraft** (Maji et al., 2013) (100 aircraft variants). We follow the standard GCD split (Vaze et al., 2022a): half of the classes are designated as known $\mathcal{Y}_l$, and 50% of the images from each known class

form the labelled set $\mathcal{D}_l$; all remaining images, from both known and novel classes, constitute the unlabelled set $\mathcal{D}_u$. A summary of the three datasets is given in Table 1.

Table 1: **Overview of datasets.** $|\mathcal{Y}_l|, |\mathcal{Y}_u|$ denote the number of classes in the labelled and unlabelled sets, and $|\mathcal{D}_l|, |\mathcal{D}_u|$ the corresponding number of images. *Balance* indicates whether the class distribution is balanced.

| Dataset | Balance | $|\mathcal{D}_l|$ | $|\mathcal{Y}_l|$ | $|\mathcal{D}_u|$ | $|\mathcal{Y}_u|$ |
|---|---|---|---|---|---|
| CUB-200-2011 (Wah et al., 2011) | ✓ | 1.5K | 100 | 4.5K | 200 |
| Stanford Cars (Krause et al., 2013) | ✓ | 2.0K | 98 | 6.1K | 196 |
| FGVC-Aircraft (Maji et al., 2013) | ✓ | 1.7K | 50 | 5.0K | 100 |

**Evaluation metrics.** We follow the standard GCD evaluation protocol (Vaze et al., 2022a). At test time, predictions on the unlabelled set are matched to ground-truth labels via the Hungarian algorithm, which resolves the unknown correspondence between predicted and true novel classes. We then report classification accuracy (ACC), defined as

$$\text{ACC} \;=\; \frac{1}{|\mathcal{D}_u|} \sum_{i=1}^{|\mathcal{D}_u|} \mathbb{1}\!\big( y_i = \mathbf{h}(\hat{y}_i) \big), \tag{15}$$

where $\hat{y}_i$ is the prediction for sample $x_i$, $y_i$ is its ground-truth label, and $\mathbf{h}$ is the optimal one-to-one mapping between predicted and ground-truth class indices obtained by the Hungarian algorithm. We report ACC on three subsets of the unlabelled set: **All** (the full unlabelled set $\mathcal{D}_u$), **Old** (unlabelled images from known classes $\mathcal{Y}_l$), and **New** (unlabelled images from novel classes $\mathcal{Y}_n = \mathcal{Y}_u \backslash \mathcal{Y}_l$). Reporting Old and New separately is important since GCD methods can trade off accuracy across the two splits, and the All metric alone hides such imbalances. In line with prior hierarchical GCD work, ACC is computed at the finest species granularity: the Hungarian matching and the All/Old/New splits are derived from the species-level predictions, while the coarser order/family heads enter only as auxiliary supervision through the cross-granularity consistency loss and are not scored as separate evaluation targets.

**Implementation details.** We use a DINOv2 ViT-B/14 (Oquab et al., 2023) backbone and fine-tune only the last transformer block, following common GCD practice (He et al., 2025; Liu & Han, 2025). The MLP projection head produces $H$ per-level features of dimension $d' = 768$, all $\ell_2$-normalised before similarity computation. We train for 200 epochs on a single NVIDIA RTX A6000 GPU with batch size $B = 128$, using AdamW with a cosine learning rate schedule. The taxonomic-similarity weights are $(\alpha, \beta, \gamma) = (0.6, 0.3, 0.1)$ on three-level datasets (CUB and FGVC-Aircraft), and analogous values on two-level Stanford Cars. For the diffusion similarity, we use $t = 2$ steps with bandwidths $\sigma_W, \sigma_D$ set to the median pairwise distance inside the mini-batch, and a hybrid blend coefficient $\lambda = 0.5$. The Wasserstein consistency weight is set to $\nu = 0.1$, which approximately matches the magnitude of the contrastive and supervised terms; we discuss the sensitivity of $\nu$ in Section 5.3. For transparency, we note that most of these knobs are inherited unchanged from the SEAL baseline: the taxonomic weights $(\alpha, \beta, \gamma)$, the consistency weight $\nu$, and the supervised/entropy weights. GeoGCD introduces only the diffusion-similarity hyperparameters $(\lambda, t, \sigma_W, \sigma_D)$ and the choice of cross-granularity divergence. Of the new hyperparameters, $\sigma_W$ and $\sigma_D$ are set automatically to the median batch distance and are not tuned, leaving $\lambda$ and $t$ as the only manually selected additions; we report a full sensitivity sweep over $\lambda$, $t$ and $\sigma_W$ in Section B and observe robust behaviour across wide ranges.

### 5.2 Main Results

Table 2 compares GeoGCD against representative GCD methods on the SSB benchmark with the DINOv2 ViT-B/14 backbone. GeoGCD obtains the best All accuracy on CUB and Stanford Cars, improving the SEAL baseline by +3.23 All on CUB and +2.40 All on Stanford Cars. Moreover, the significant result achieves **87.73** on CUB with Old setting, higher than the SOTA DebGCD by **6.93**. It also attains the best Old accuracy on all three datasets, including 80.46 on FGVC-Aircraft.

Table 2: **Comparison of GCD methods on the SSB benchmark** (Vaze et al., 2022b) with the DINOv2 ViT-B/14 backbone. Accuracy (%) is reported on the All / Old / New splits; the rightmost column is the average All across the three datasets. The highest and second-highest scores in each column are in **bold** and underlined. Results for prior methods are taken from (He et al., 2025). Our results use a fixed seed; the mean$_{\pm std}$ over three seeds is reported in Appendix C.3 and preserves the relative ordering.

| Method | Venue | CUB | | | Stanford Cars | | | FGVC-Aircraft | | | Avg. |
| | | All | Old | New | All | Old | New | All | Old | New | All |
|---|---|---|---|---|---|---|---|---|---|---|---|
| $k$-means (MacQueen, 1967) | – | 67.6 | 60.6 | 71.1 | 29.4 | 24.5 | 31.8 | 18.9 | 16.9 | 19.9 | 38.6 |
| GCD (Vaze et al., 2022a) | CVPR'22 | 71.9 | 71.2 | 72.3 | 65.7 | 67.8 | 64.7 | 55.4 | 47.9 | 59.2 | 64.3 |
| CiPR (Hao et al., 2023) | TMLR'24 | 78.3 | 73.4 | **80.8** | 66.7 | 77.0 | 61.8 | 59.2 | 65.0 | 56.3 | 68.1 |
| SimGCD (Wen et al., 2023) | ICCV'23 | 71.5 | 78.1 | 68.3 | 71.5 | 81.9 | 66.6 | 63.9 | 69.9 | 60.9 | 69.0 |
| $\mu$GCD (Vaze et al., 2023) | NeurIPS'23 | 74.0 | 75.9 | 73.1 | 76.1 | 91.0 | 68.9 | 66.3 | 68.7 | 65.1 | 72.1 |
| SPTNet (Wang et al., 2024b) | ICLR'24 | 76.3 | 79.5 | 74.6 | – | – | – | – | – | – | – |
| DebGCD (Liu & Han, 2025) | ICLR'25 | 77.5 | 80.8 | 75.8 | 75.4 | 87.7 | 69.5 | 71.9 | 76.0 | 69.8 | 74.9 |
| SEAL (He et al., 2025) | NeurIPS'25 | 76.7 | 78.3 | 75.9 | 77.7 | 88.7 | **72.4** | **74.6** | 73.2 | **75.3** | 76.3 |
| **GeoGCD (Ours)** | – | **79.93** | **87.73** | 72.20 | **80.10** | **93.57** | 67.11 | 69.85 | **80.46** | 59.22 | **76.6** |

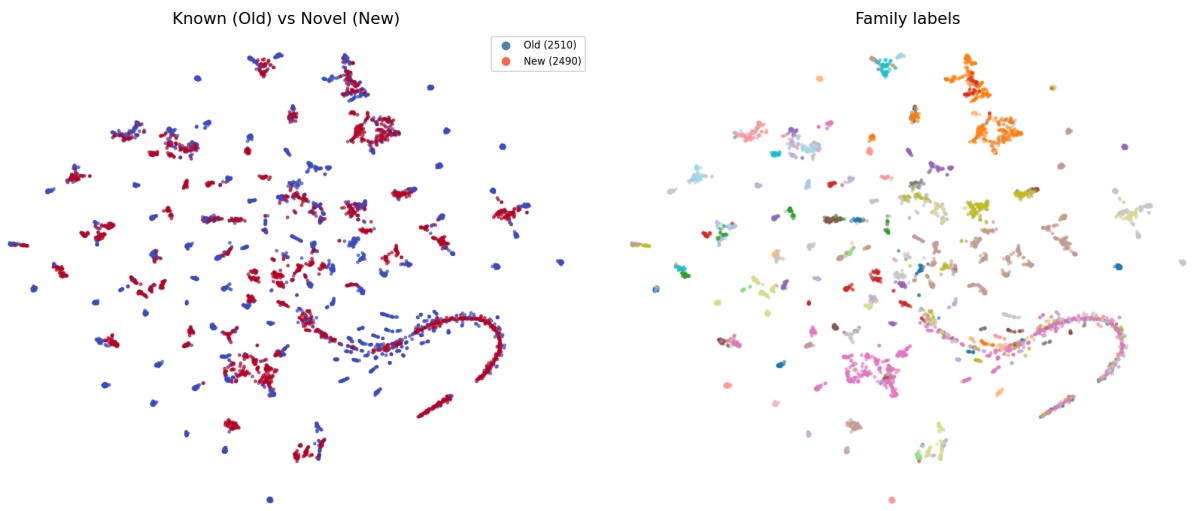

Figure 2: **t-SNE of GeoGCD species-level projections on CUB.** *Left:* known (Old, blue) and novel (New, red) samples both form tight, well-separated clusters. *Right:* the same embedding coloured by family; samples of the same family stay close, showing that GeoGCD preserves the taxonomic geometry while still discovering novel classes.

On CUB and Stanford Cars, GeoGCD attains the best overall (All) accuracy among all compared methods. These gains are driven by a large improvement on the **Old** split (+9.4 and +4.9 over SEAL), reflecting better preservation of the global semantic geometry for known classes. The improvement comes with a trade-off on the **New** split, where GeoGCD trails the strongest baselines by 3.6–5.3 % ; on FGVC-Aircraft this trade-off is more pronounced and the overall accuracy regresses, a failure mode we analyse in Section 5.3. On CUB, GeoGCD reaches 79.93 All / 87.73 Old / 72.20 New, and on Stanford Cars 80.10 All / 93.57 Old / 67.11 New. The All accuracy gains on these two datasets, despite their very different visual statistics, indicate that the geometric signals introduced by GeoGCD strengthen known-class structure across domains; whether this transfers to novel classes depends on how well the taxonomy aligns with the data manifold, as the Aircraft result shows.

## 5.3 Ablation Study

We isolate the contribution of each module on CUB and Stanford Cars by progressively adding the diffusion-based similarity (**+ Diff**) and the Wasserstein cross-granularity loss (**+ W**) to the soft contrastive baseline that uses taxonomic similarity and KL consistency. Results are reported in Table 3.

Table 3: **Ablation on CUB-200-2011, Stanford Cars and FGVC-Aircraft** Starting from the soft contrastive baseline with taxonomic similarity and KL consistency, we progressively add the diffusion-based similarity to the contrastive supervision and replace the KL term by the 1-Wasserstein objective. Single seed for a controlled comparison; see Table 11 for variance.

| Variant | CUB-200-2011 | | | Stanford Cars | | | FGVC-Aircraft | | |
|---|---|---|---|---|---|---|---|---|---|
| | **All** | **Old** | **New** | **All** | **Old** | **New** | **All** | **Old** | **New** |
| Baseline (taxonomic + KL) | 76.70 | 78.30 | **75.90** | 77.70 | 88.70 | **72.40** | **74.60** | 73.20 | **75.30** |
| + Diff | 76.44 | 87.59 | 65.40 | 78.47 | 93.47 | 64.01 | 73.51 | **81.77** | 65.23 |
| + Diff + W (**GeoGCD**) | **79.93** | **87.73** | 72.20 | **80.10** | **93.57** | 67.11 | 69.01 | 79.92 | 58.08 |

**Effect of the diffusion similarity.** Adding the diffusion-based signal to the taxonomic similarity (*+ Diff*) substantially improves Old accuracy on both datasets (+9.29 on CUB, +4.77 on Stanford Cars), which indicates that the manifold-aware similarity helps the model exploit the cluster structure of known classes more effectively: the random-walk distances on the batch graph capture density modes that the taxonomy tree cannot directly express, sharpening intra-cluster similarity for known classes. The downside, however, is consistent across both datasets: New accuracy drops compared to the baseline (−10.50 on CUB, −8.39 on Stanford Cars), which suggests that the diffusion signal alone, when paired with the standard KL consistency, can over-commit the representation to the known-class geometry: novel classes are still poorly clustered early in training, so the data-driven similarity inherits this noise and pushes the consistency loss towards reinforcing known-class structure rather than discovering new modes.

**Effect of the Wasserstein consistency.** Replacing the KL term by the 1-Wasserstein objective (*+ Diff + W*) yields the best result on every metric across both CUB and Stanford Cars. Compared to the baseline, All accuracy improves by +3.23 on CUB and +2.40 on Stanford Cars; compared to *+ Diff*, the gain on the New split is +6.80 on CUB and +3.10 on Stanford Cars. On Stanford Cars, the Wasserstein objective also slightly improves Old accuracy further over *+ Diff* (93.47 → 93.57), showing that the alignment signal does not trade off known-class accuracy when stabilised this way. The recovery on the New split is particularly meaningful: it shows that swapping KL for $W_1$ does not merely stabilise training, but specifically restores the ability of the cross-level alignment to benefit novel classes, even when the upstream contrastive signal is biased towards known-class geometry. This pattern is consistent with the disjoint-support analysis of Lemma 1 and Propositions 1 and 2: when predictions over novel classes are noisy and the predicted and inferred coarse distributions concentrate on nearly disjoint regions of the label simplex, KL is either undefined or carries gradients that ignore the spatial separation of supports, while $W_1$ remains bounded and provides a non-vanishing gradient that shrinks the geometric distance between the two distributions. The Wasserstein term thus complements the diffusion similarity by allowing the cross-level alignment signal to remain informative exactly in the regime where novel classes are most fragile.

**When taxonomy and manifold disagree: FGVC-Aircraft.** FGVC-Aircraft is the one benchmark where this mechanism does not hold. On Aircraft, the diffusion signal behaves as on the other datasets (+8.57 Old, −10.07 New over the baseline), but adding the Wasserstein consistency worsens rather than recovers novel classes (65.23 → 58.08 New, 73.51 → 69.01 All), and the full model regresses below the baseline on All and New. We attribute this to a mismatch between the Aircraft taxonomy and the visual manifold: its coarse groupings (manufacturer/family) are only weakly predictive of appearance, since visually near-identical variants can belong to different families and visually distinct models can share one. When the coarse target is itself unreliable, enforcing cross-granularity agreement propagates an inconsistent signal,

and the geometry-aware consistency term amplifies it rather than correcting it. This is the failure mode anticipated in our setup: the benefit of geometry-guided alignment is contingent on the taxonomy being approximately consistent with the data manifold. The results in Section B confirms that the instability is tied to how poorly the coarse structure constrains the fine predictions on this dataset rather than to the metric alone.

**Sensitivity to the consistency weight.** The Wasserstein term has a different magnitude than KL, so its loss weight $\nu$ has to be rescaled accordingly. Setting $\nu = 1.0$ destabilises training; rescaling to $\nu = 0.1$, which approximately matches the contrastive-term magnitude, recovers stable training and yields the gains reported above. A full sensitivity sweep over $\nu$ is provided in the appendix.

**Summary.** The two contributions are complementary: the diffusion similarity strengthens the representation of known classes by injecting data-driven manifold structure, while the Wasserstein consistency ensures that this geometric signal propagates correctly across granularities, especially on novel classes whose coarse-level distributions are still poorly aligned with the pseudo-coarse ones. Their combination yields the best result on every reported split across both CUB-200-2011 and Stanford Cars.

## 6 Conclusion

We presented GeoGCD, a geometry-guided hierarchical framework for Generalized Category Discovery built around the principle that the global geometry of semantics, the shape that taxonomic relations carve in feature and label space, should be a first-class learning target rather than an auxiliary signal. The framework realises this principle through two components: a hybrid soft-label matrix that fuses taxonomic similarity with a manifold-aware diffusion similarity for hierarchical contrastive learning, and a Wasserstein cross-granularity consistency loss that aligns predictions across levels with formal guarantees of finiteness, continuity, and gradient informativeness under disjoint support, the regime that arises with overwhelming probability at random initialisation. Empirically, GeoGCD improves overall and, most consistently, known-class (Old) accuracy on CUB and Stanford Cars, at the cost of a trade-off on the novel-class split; on FGVC-Aircraft, where the taxonomy is poorly aligned with the visual manifold, this trade-off turns into a regression, which we analyse explicitly. Sensitivity analysis further shows that the geometric framework, rather than careful hyperparameter tuning, is what drives these improvements.

**Limitations.** Three limitations of the present work are worth noting. First, our theoretical analysis of the 1-Wasserstein term characterises the disjoint-support regime that is typical at random initialisation, but does not provide a tight description of the intermediate phase of training where the predicted and pseudo-coarse distributions begin to overlap. Second, GeoGCD assumes that a curated semantic taxonomy is available at training time and is not designed for settings where the hierarchy itself must be inferred jointly with the representation. Third, the diffusion similarity is computed within each mini-batch, so the random-walk structure that informs the hybrid soft-label matrix is local relative to the batch size; extending the construction to a global graph defined over a memory bank could strengthen the manifold-aware component at the cost of additional bookkeeping.

### Acknowledgments

TODO: Acknowledgments (hidden in anonymous submission).

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

## A  Analysis on Generic Datasets

To assess whether the geometric framework of GeoGCD generalises beyond the standard fine-grained natural-image benchmarks, we evaluate it on two domain-specific datasets that exhibit very different visual statistics from CUB or Stanford Cars: a clinical retinal imaging dataset with strong class imbalance and an ordinal severity structure, and a histopathology dataset with balanced texture-like classes. Unlike the SSB benchmarks, both datasets exhibit a shallow 2-level taxonomy: a single domain-level root grouping the leaf classes, with no intermediate family structure. This setting tests whether GeoGCD remains effective when the hierarchical prior degenerates and the diffusion similarity has to carry most of the structural signal.

**Datasets.**  **APTOS 2019** (Aftab & Akhtar, 2025) is a diabetic retinopathy (DR) grading benchmark of 3,662 retinal fundus images annotated with five severity grades (No DR, Mild, Moderate, Severe, Proliferative). The hierarchy is shallow: all five grades share a common root *DR Severity*. We treat the three earliest grades (No DR, Mild, Moderate; 3,174 images, 86.7%) as known classes $\mathcal{Y}_l$ and the two most severe grades (Severe, Proliferative; 488 images, 13.3%) as novel classes $\mathcal{Y}_n$. The dataset poses two distinctive challenges relative to the SSB benchmark: a strong class imbalance and an ordinal label space, where adjacent severity grades are visually similar.

**NCT-CRC-HE-100K** (Kather et al., 2018) is a colorectal histopathology dataset of 100,000 patches at $224 \times 224$ resolution, annotated into nine tissue types (adipose, background, debris, lymphocytes, mucus, smooth muscle, normal mucosa, stroma, tumour). All nine classes share a single root *Tissue*. We designate the first five classes (ADI, BACK, DEB, LYM, MUC; 52,938 images, 52.9%) as known and the remaining four (MUS, NORM, STR, TUM; 47,062 images, 47.1%) as novel. Compared to APTOS, the class distribution is approximately balanced (each class contains between 8.7K and 14.3K images) and the visual content is texture-dominated, which is closer to the regime in which diffusion-based similarity is expected to be informative. A class-level breakdown of both datasets is given in Table 4.

**Results.**  Table 5 reports the best All accuracy obtained by GeoGCD with and without the Wasserstein cross-granularity term, together with a no-diffusion baseline on NCT-CRC-HE-100K.

**Discussion.**  On NCT-CRC-HE-100K, GeoGCD attains the highest accuracy on both the All and New splits (95.32 All / 90.38 New), improving the no-diffusion baseline by +1.89 All and the diffusion-only variant by +0.42 All, with consistent gains on novel classes (+3.94 over no-diffusion). The Old accuracy remains essentially saturated above 99.6 across all variants, indicating that the additional geometric signal does not harm known-class performance even when the latter is already very strong. The fact that GeoGCD continues to improve under the shallow *Tissue* root, where the taxonomic prior carries no discriminative information beyond a single grouping, confirms that the manifold-aware diffusion signal is the main contributor in this regime, and that the Wasserstein term propagates this signal across granularities without destabilising training.

On APTOS 2019, the diffusion-only variant achieves the best overall and known-class accuracy (78.62 All, 84.72 Old), while GeoGCD with the Wasserstein term recovers a notably higher accuracy on novel classes

Table 4: **Class-level overview of the generic datasets.** For each class we report the total image count and its split into labelled ($\mathcal{D}_l$) and unlabelled ($\mathcal{D}_u$) sets. Known classes contribute 80% of their images to $\mathcal{D}_l$ and the remaining 20% to $\mathcal{D}_u$; novel classes contribute their entire content to $\mathcal{D}_u$. The full unlabelled pool is used for evaluation.

| Cls | Name | Total | $|\mathcal{D}_l|$ | $|\mathcal{D}_u|$ | Type |
|---|---|---|---|---|---|
| *APTOS 2019* | | | | | |
| 0 | No DR | 1,805 | 1,444 | 361 | Known |
| 1 | Mild | 370 | 296 | 74 | Known |
| 2 | Moderate | 999 | 799 | 200 | Known |
| 3 | Severe | 193 | 0 | 193 | Novel |
| 4 | Proliferative | 295 | 0 | 295 | Novel |
| | **Total** | 3,662 | 2,539 | 1,123 | |
| *NCT-CRC-HE-100K* | | | | | |
| 0 | ADI (adipose) | 10,407 | 8,326 | 2,081 | Known |
| 1 | BACK (background) | 10,566 | 8,453 | 2,113 | Known |
| 2 | DEB (debris) | 11,512 | 9,210 | 2,302 | Known |
| 3 | LYM (lymphocytes) | 11,557 | 9,246 | 2,311 | Known |
| 4 | MUC (mucus) | 8,896 | 7,117 | 1,779 | Known |
| 5 | MUS (smooth muscle) | 13,536 | 0 | 13,536 | Novel |
| 6 | NORM (normal mucosa) | 8,763 | 0 | 8,763 | Novel |
| 7 | STR (stroma) | 10,446 | 0 | 10,446 | Novel |
| 8 | TUM (tumour) | 14,317 | 0 | 14,317 | Novel |
| | **Total** | 100,000 | 42,352 | 57,648 | |

Table 5: **Results on generic datasets.** "Diffusion" refers to the soft contrastive baseline with hybrid diffusion-taxonomic similarity and KL cross-granularity consistency; "Diff+W" is GeoGCD with the 1-Wasserstein consistency replacing KL; "No Diff" uses purely taxonomic similarity. Best All in **bold**.

| Dataset | Variant | All | Old | New |
|---|---|---|---|---|
| APTOS 2019 | Diffusion | **78.62** | **84.72** | 38.93 |
| | Diff+W (**GeoGCD**) | 76.73 | 81.76 | **44.06** |
| NCT-CRC-HE-100K | No Diff | 93.43 | 99.64 | 86.44 |
| | Diffusion | 94.90 | **99.78** | 89.41 |
| | Diff+W (**GeoGCD**) | **95.32** | 99.71 | **90.38** |

(44.06 vs. 38.93, +5.13). The reversed ranking on the All metric is a direct consequence of the strong class imbalance (86.7% known): a gain on the small novel set (13.3%) is more than compensated by a small drop on the dominant known set when predictions are summed over $\mathcal{D}_u$. Importantly, the trend on the New split, the regime that category discovery is ultimately concerned with, follows the same direction as on the SSB benchmarks: replacing KL by $W_1$ improves novel-class performance, which is consistent with the disjoint-support analysis of Lemma 1 and Propositions 1 and 2, and is particularly relevant on the ordinal label space of DR severity, where the 1-Wasserstein distance respects the natural ordering of severity grades.

# B   Analysis on Multiple Settings

We conduct sensitivity analysis on three key hyperparameters of GeoGCD: the hybrid blend coefficient $\lambda$ controlling the balance between taxonomic and diffusion similarities, the number of diffusion steps $t$, and the Gaussian kernel bandwidth $\sigma_W$ used in the diffusion graph. All ablations are run on CUB-200-2011, Stanford

Table 6: **Sensitivity to the hybrid blend coefficient $\lambda$.** $\lambda = 0$ uses only the diffusion similarity while $\lambda = 1$ uses only the taxonomic similarity; the default value $\lambda = 0.5$ is reported in Table 2. Best results per dataset in **bold**.

| $\lambda$ | CUB-200-2011 | | | Stanford Cars | | | FGVC-Aircraft | | |
|---|---|---|---|---|---|---|---|---|---|
| | All | Old | New | All | Old | New | All | Old | New |
| 0.00 | **78.87** | **86.55** | **71.27** | 80.05 | **92.50** | 68.04 | **77.14** | 81.12 | **73.15** |
| 0.25 | 77.11 | 85.92 | 68.38 | – | – | – | 75.94 | 81.18 | 70.69 |
| 0.75 | 77.82 | 85.75 | 69.97 | **80.34** | 92.00 | **69.09** | 69.97 | 80.28 | 59.64 |
| 1.00 | 77.93 | 86.03 | 69.90 | 79.49 | **92.50** | 66.94 | 71.41 | **81.35** | 61.44 |

Table 7: **Sensitivity to the number of diffusion steps $t$.** The default value $t = 2$ is reported in Table 2. Best results per dataset in **bold**.

| $t$ | CUB-200-2011 | | | Stanford Cars | | | FGVC-Aircraft | | |
|---|---|---|---|---|---|---|---|---|---|
| | All | Old | New | All | Old | New | All | Old | New |
| 1 | **79.72** | **87.07** | **72.44** | **80.84** | **94.55** | **67.60** | 74.68 | 78.72 | **70.63** |
| 3 | 78.79 | 86.79 | 70.86 | – | – | – | **74.83** | **79.08** | 70.57 |
| 5 | 78.65 | 86.37 | 71.00 | 79.59 | 92.98 | 66.67 | 71.86 | 80.22 | 63.48 |

Cars, and FGVC-Aircraft with the default configuration ($\lambda = 0.5$, $t = 2$, $\sigma_W = 1.0$) as the reference; results are reported in Tables 6, 7, and 8.

**Effect of the hybrid blend $\lambda$.** Table 6 shows that the optimal $\lambda$ depends on the visual structure of the dataset. On CUB, $\lambda = 0$ (pure diffusion) achieves the best result (78.87 All), with the gap to other values around 1–1.8 points; this suggests that for fine-grained biological categories with elongated, density-varying clusters, the manifold-aware diffusion signal is sufficient and the taxonomic prior is partially redundant. On Stanford Cars, the trend reverses: $\lambda = 0.75$ (heavy taxonomic weight) yields the best result (80.34 All), indicating that the rigid visual differences across car models are better captured by the taxonomy than by the feature manifold. On Aircraft, $\lambda = 0$ again outperforms larger values, similar to CUB. Overall, $\lambda \in [0, 0.75]$ is a safe range, and the choice within this range adapts to the geometry of each domain.

**Effect of the diffusion steps $t$.** Table 7 reveals a consistent preference for $t = 1$ across CUB and Stanford Cars. On CUB, $t = 1$ achieves 79.72 All, decreasing monotonically as $t$ grows; on Stanford Cars, $t = 1$ similarly dominates (80.84 All vs. 79.59 at $t = 5$). This is consistent with the spectral interpretation of diffusion maps (Section 3.3): larger $t$ progressively erases fine local structure as the random walk approaches the stationary distribution, weakening the discriminative signal that the contrastive loss relies on. On Aircraft, $t = 1$ and $t = 3$ are comparable, with $t = 5$ clearly worse, supporting the same conclusion.

**Effect of the Gaussian kernel bandwidth $\sigma_W$.** Table 8 shows that GeoGCD is robust across $\sigma_W \in \{0.5, 2.0, 5.0\}$ on CUB, with All accuracy varying within ~1.5 points. The best result is obtained at $\sigma_W = 5.0$ (79.43 All), although the smaller $\sigma_W = 0.5$ gives the highest New accuracy (72.99). This suggests a small trade-off between intra-cluster sharpness (small $\sigma_W$, favouring novel-class discovery) and global connectivity (larger $\sigma_W$, favouring known-class consolidation), and indicates that the median-distance heuristic used in our default configuration is a reasonable compromise.

**Summary.** The three hyperparameters control different aspects of the geometric signal: $\lambda$ balances prior versus data-driven similarity, $t$ controls the locality scale of the random walk, and $\sigma_W$ governs the connectivity of the batch graph. Across all three, GeoGCD remains within a narrow performance band, with no single setting dominating in absolute terms. This stability suggests that the geometric framework itself, rather than careful hyperparameter tuning, is what drives the improvements reported in Section 5.2.

Table 8: **Sensitivity to the Gaussian kernel bandwidth $\sigma_W$ on CUB-200-2011.** The default value $\sigma_W = 1.0$ is reported in Table 2. Best results in **bold**.

| $\sigma_W$ | **All** | **Old** | **New** |
|---|---|---|---|
| 0.5 | 78.75 | 84.57 | **72.99** |
| 2.0 | 77.89 | 85.40 | 70.45 |
| 5.0 | **79.43** | **86.23** | 72.68 |

## C  Robustness of the Cross-Granularity Distance

This appendix studies how the choice of distance in the cross-granularity consistency term affects performance, complementing the main ablation (Table 3). All runs use the default GeoGCD configuration and the same seed; accuracy is the best-epoch ACC on the test split, as in the main results.

### C.1  Experiments with Additional Distances

Our consistency term uses the 1-Wasserstein distance, but the property we rely on – finiteness and an informative gradient under disjoint support – is also shared by other bounded divergences. To isolate the role of the specific distance, we replace $W_1$ by three bounded, permutation-invariant alternatives: total variation (TV), Hellinger, and Jensen–Shannon (JS). Table 9 reports the result.

Table 9: **Additional cross-granularity distances.** Best-epoch test accuracy (%) on the All/Old/New splits, GeoGCD default config, single seed for a controlled comparison (Table 11).

| Distance | CUB-200-2011 | | | Stanford Cars | | | FGVC-Aircraft | | |
|---|---|---|---|---|---|---|---|---|---|
| | All | Old | New | All | Old | New | All | Old | New |
| $W_1$ (ours) | 79.93 | **87.73** | 72.20 | 80.10 | **93.57** | 67.11 | 69.85 | **80.46** | 59.22 |
| TV | 79.10 | 84.92 | 73.33 | 79.90 | 92.48 | 67.77 | 73.18 | 79.86 | 66.49 |
| Hellinger | **81.38** | 84.50 | **78.28** | 79.84 | 92.27 | 67.85 | 73.99 | 78.36 | 69.61 |
| JS | 80.07 | 85.47 | 74.71 | **80.97** | 91.16 | **71.15** | **75.07** | 78.36 | **71.77** |

Three observations follow. **(i) Bounded distances improve novel-class accuracy.** On every dataset, TV/Hellinger/JS raise New over $W_1$ (e.g. Hellinger +6.1 on CUB, JS +4.0 on Stanford Cars), indicating that a permutation-invariant target is gentler on the uncertain novel-class predictions. **(ii) Wasserstein preserves known-class structure best.** On all three datasets $W_1$ attains the highest Old accuracy (87.73/93.57/80.46), which suggests that its transport geometry is the most compatible with our diffusion-map similarity: aligning coarse predictions through a ground-metric-aware distance reinforces exactly the known-class geometry that the diffusion signal sharpens. **(iii) Overall accuracy is comparable where the taxonomy is geometrically faithful.** On CUB and Stanford Cars, All differs by less than 1.5 points across all four distances, so the choice mainly trades Old against New. FGVC-Aircraft is the exception – there the bounded distances also lead on All – which is consistent with the taxonomy–manifold mismatch discussed in Section 5.3: when the coarse taxonomy is weakly predictive of appearance, the geometry coupling of $W_1$ no longer pays off and a simpler permutation-invariant distance is preferable. In short, $W_1$ is the strongest choice precisely when the semantic geometry it exploits is reliable, and it remains our default for that reason; the alternatives are a practical option when the taxonomy is known to be loosely aligned with the data.

### C.2  Permutation Invariance of the Consistency Term

Because the closed-form $W_1$ uses the cumulative distribution over coarse indices, its value depends in principle on the ordering of the coarse classes, which is arbitrary for non-ordinal taxonomies. We probe sensitivity to this ordering by re-training with fixed random permutations of the coarse-class columns (Table 10). On CUB and Stanford Cars the results are essentially unchanged – All stays within about one point across permutations – so the reported gains do not rely on any particular ordering. On FGVC-Aircraft the results

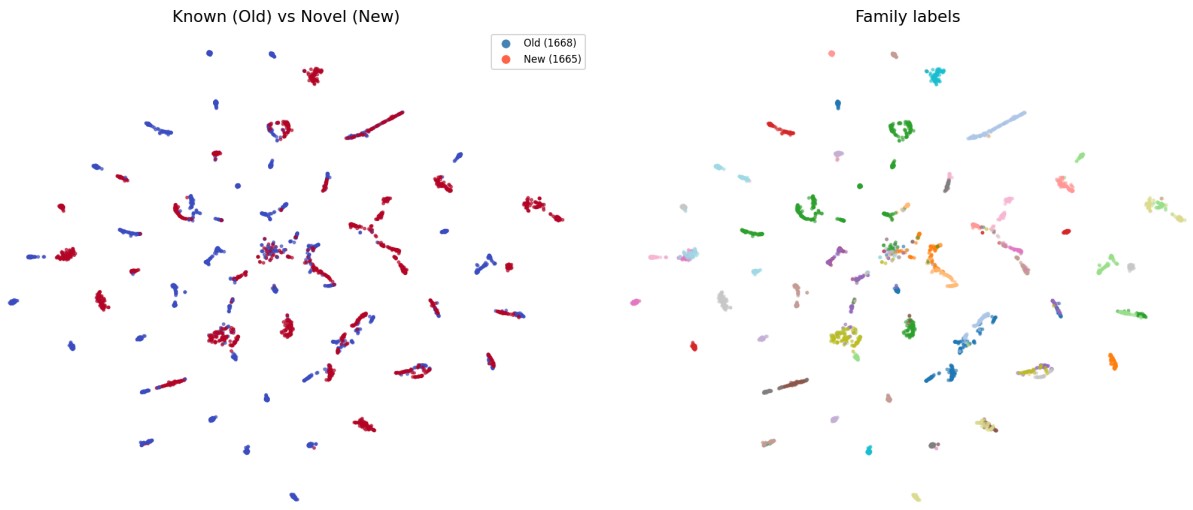

Figure 3: **t-SNE of GeoGCD ($W_1$) species-level projections on FGVC-Aircraft.** Unlike CUB (Figure 2), the family colouring (*right*) is fragmented and interleaved: samples of the same family are scattered across many clusters. This visualises the taxonomy–manifold mismatch that makes the geometry-coupled consistency term less effective on this dataset.

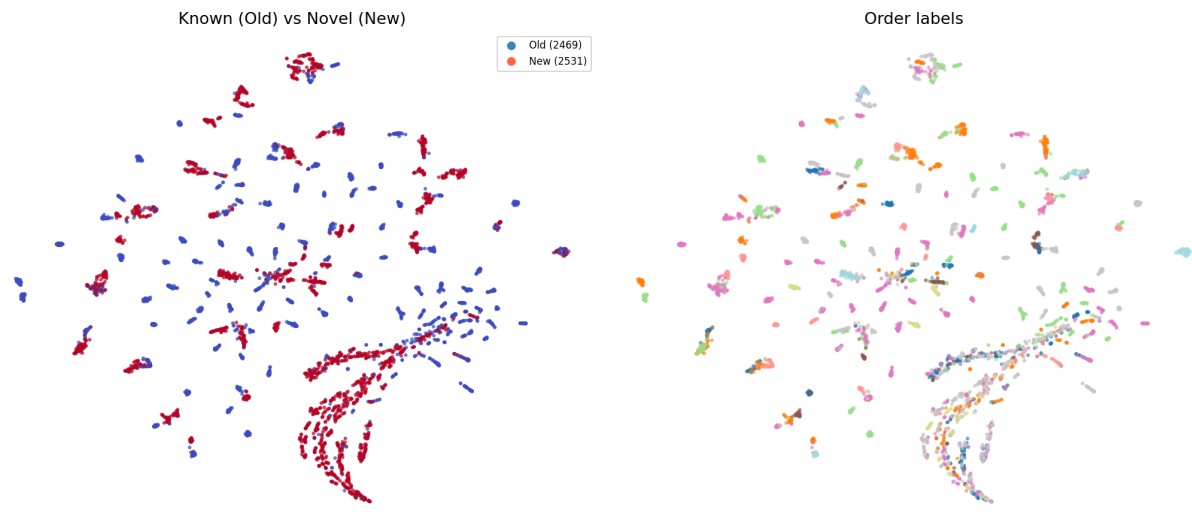

Figure 4: **t-SNE of GeoGCD ($W_1$) on Stanford Cars.** As on CUB, known (Old) and novel (New) classes form tight, well-separated clusters, and same-make (coarse) samples stay close.

vary more (All from 69.9 to 74.4), once again matching the mismatch analysis: when the taxonomy is weakly aligned with the manifold, the consistency term is more sensitive to how the coarse classes are arranged.

## C.3 Variance across Seeds

Table 11 reports GeoGCD ($W_1$) and the JS variant over three random seeds; the main-paper Table 2 reports the fixed-seed run, and this table gives the corresponding mean$_{\pm\text{std}}$. Two points are worth noting. First, Stanford Cars is very stable (All std $\leq$ 0.7), whereas CUB and FGVC-Aircraft show larger run-to-run

Table 10: **Sensitivity to coarse-class ordering.** Best-epoch test accuracy (%) of GeoGCD ($W_1$) under the original ordering and two fixed random permutations of the coarse-class columns.

| Ordering | CUB-200-2011 | | | Stanford Cars | | | FGVC-Aircraft | | |
|---|---|---|---|---|---|---|---|---|---|
| | All | Old | New | All | Old | New | All | Old | New |
| Original | 79.93 | 87.73 | 72.20 | 80.10 | 93.57 | 67.11 | 69.85 | 80.46 | 59.22 |
| Permutation 1 | 80.24 | 87.55 | 72.99 | 80.18 | 92.88 | 67.92 | 72.58 | 79.74 | 65.41 |
| Permutation 2 | 79.29 | 85.02 | 73.61 | 80.40 | 93.01 | 68.24 | 74.44 | 80.10 | 68.77 |

variation (All std up to $\sim 2$), so single-seed numbers on those two datasets should be read with a margin of about two points. Second, the qualitative conclusions of this paper are stable across seeds: $W_1$ retains the best Old accuracy, the JS variant retains the best New accuracy, and the FGVC-Aircraft gap between the two persists. The novel-class trade-off relative to the strongest baselines is therefore a genuine effect rather than seed noise, which is why we report it transparently.

Table 11: **Variance across seeds.** Best-epoch test accuracy (%), mean$_{\pm \text{std}}$ over seeds, for the default GeoGCD ($W_1$) and the JS variant.

| Method | CUB-200-2011 | | | Stanford Cars | | | FGVC-Aircraft | | |
|---|---|---|---|---|---|---|---|---|---|
| | All | Old | New | All | Old | New | All | Old | New |
| GeoGCD ($W_1$) | $78.1_{\pm 1.3}$ | $85.3_{\pm 2.4}$ | $70.9_{\pm 1.9}$ | $79.8_{\pm 0.5}$ | $93.6_{\pm 0.3}$ | $66.5_{\pm 1.4}$ | $70.0_{\pm 2.0}$ | $80.6_{\pm 1.8}$ | $59.4_{\pm 2.1}$ |
| JS | $78.2_{\pm 1.3}$ | $83.8_{\pm 1.4}$ | $72.7_{\pm 1.6}$ | $80.0_{\pm 0.7}$ | $91.3_{\pm 0.1}$ | $69.0_{\pm 1.5}$ | $75.0_{\pm 0.8}$ | $78.9_{\pm 1.9}$ | $71.1_{\pm 0.5}$ |

### C.4 Deferred Proofs

This appendix collects the proofs of the formal statements in Section 4.5 (Lemma 1 and Propositions 1 and 2), deferred from the main text.

*Proof of Lemma 1.* For the single-mode case, the two predicted indices coincide with probability $1/K_h$ (uniform independent draws), so they are disjoint with probability $1 - 1/K_h$. For the $s$-mode case, the expected size of the intersection of two uniformly chosen size-$s$ subsets of $\{1, \ldots, K_h\}$ is $s^2/K_h$, so by Markov's inequality the probability that the intersection is non-empty is at most $s^2/K_h$, which gives the stated bound. $\square$

*Proof of Proposition 1.* **(1)** For any coupling $\gamma \in \Pi(p, q)$, $\sum_{k,k'} |k - k'| \gamma(k, k') \leq (K - 1) \sum_{k,k'} \gamma(k, k') = K - 1$. Thus, $W_1(p, q) \leq K - 1$. Since the CDF form $W_1(p, q) = \sum_{k=1}^{K-1} |F_p(k) - F_q(k)|$ is a finite sum of continuous functions of $p, q$, it is continuous on the product simplex.

**(2)** Take $p_n = \delta_{k_0}$ and $q_n = \delta_{k_0 + \epsilon_n}$ with $\epsilon_n \downarrow 0$ along a refined grid (or, in the discrete case, adjacent coordinates that get closer in some refinement). Since the supports are disjoint, $\text{KL}(p_n \| q_n)$ contains a term $p_n(k_0) \log(p_n(k_0)/0) = +\infty$. CDFs $F_{p_n}, F_{q_n}$ differ only in a vanishing interval, then $W_1(p_n, q_n) \to 0$. $\square$

*Proof of Proposition 2.* **(1)** Under disjoint support, there exists $k$ with $q(k) > 0$ and $p_{\theta_0}(k) = 0$, so $\text{KL}(q \| p_{\theta_0}) = +\infty$ and the gradient is undefined. For the reverse direction, finiteness requires $p_{\theta_0}(k) = 0$ wherever $q(k) = 0$, which fails when supports are disjoint; when finite, the gradient $\partial_\theta \text{KL}(p_{\theta_0} \| q) = \sum_k (\log(p_{\theta_0}(k)/q(k)) + 1) \partial_\theta p_{\theta_0}(k)$ depends only on the ratio $p_{\theta_0}/q$ on the overlap, with no dependence on the geometric distance between the two supports.

**(2)** Differentiating $W_1(p, q) = \sum_{k=1}^{K-1} |F_p(k) - F_q(k)|$ with respect to $p(j)$ and using $\partial F_p(k)/\partial p(j) = \mathbb{1}[j \leq k]$ yields equation 14. Each surviving term contributes $\pm 1$, so the magnitude is the number of CDF disagreements, which grows with the support separation. $\square$

### C.5 Training Dynamics of the Consistency Term

The formal results in Section 4.5 characterise the disjoint-support regime that is typical at initialisation. The following proposition complements them by describing how that regime evolves during training under the Wasserstein consistency gradient.

**Proposition 3** (Transience of the disjoint-support regime under the $W_1$ update). *Fix a coarse level $h$, treat the pseudo-coarse target $\tilde{q} := \tilde{p}_i^{(h)}$ as constant, and consider minimising $\ell(p) := W_1(p, \tilde{q})$ over $p \in \Delta^{K-1}$ by (projected) gradient descent. Then* (i) *$\ell$ is convex and piecewise-linear, with $\ell(p) = 0$ if and only if $p = \tilde{q}$; and* (ii) *the subgradient $\partial\ell/\partial p(j) = \sum_{k \geq j} \mathrm{sign}(F_p(k) - F_{\tilde{q}}(k))$ of Proposition 2 moves probability mass from coarse indices where the predicted CDF runs ahead ($F_p > F_{\tilde{q}}$) to those where it lags ($F_p < F_{\tilde{q}}$), so each step transports $p$ towards $\tilde{q}$ and strictly decreases $\ell$ whenever $p \neq \tilde{q}$. Hence the disjoint-support configuration is not stationary: the supports are driven to overlap as training proceeds, i.e. the disjoint-support regime is transient under the $W_1$-CGC gradient. By contrast $\mathrm{KL}(\tilde{q} \| p) = +\infty$ with an undefined gradient while the supports are disjoint, so it offers no descent direction out of this regime and becomes informative only after overlap has been established by other terms. Thus $W_1$ governs not only the initialisation regime (Lemma 1) but also the exit from disjoint support during training.*

*Proof.* (i) $\ell(p) = \sum_{k=1}^{K-1} |F_p(k) - F_{\tilde{q}}(k)|$ is a finite sum of absolute values of affine functions of $p$, hence convex and piecewise-linear, and it vanishes iff $F_p = F_{\tilde{q}}$, i.e. $p = \tilde{q}$. (ii) The subgradient is that of Proposition 2. For a convex function, a step along $-\partial\ell$ with sufficiently small step size does not increase $\ell$ and strictly decreases it at non-stationary points; since $p = \tilde{q}$ is the unique minimiser, $\ell$ decreases monotonically towards 0 along the flow. The sign of each subgradient coordinate points in the direction that reduces the accumulated CDF gap, which is exactly the mass-transport direction stated. For KL, disjoint support means some index $k$ has $\tilde{q}(k) > 0$ and $p(k) = 0$, giving $\mathrm{KL}(\tilde{q} \| p) = +\infty$ with undefined gradient. A full coupled analysis in which $\tilde{q}$ also evolves is left to future work. □

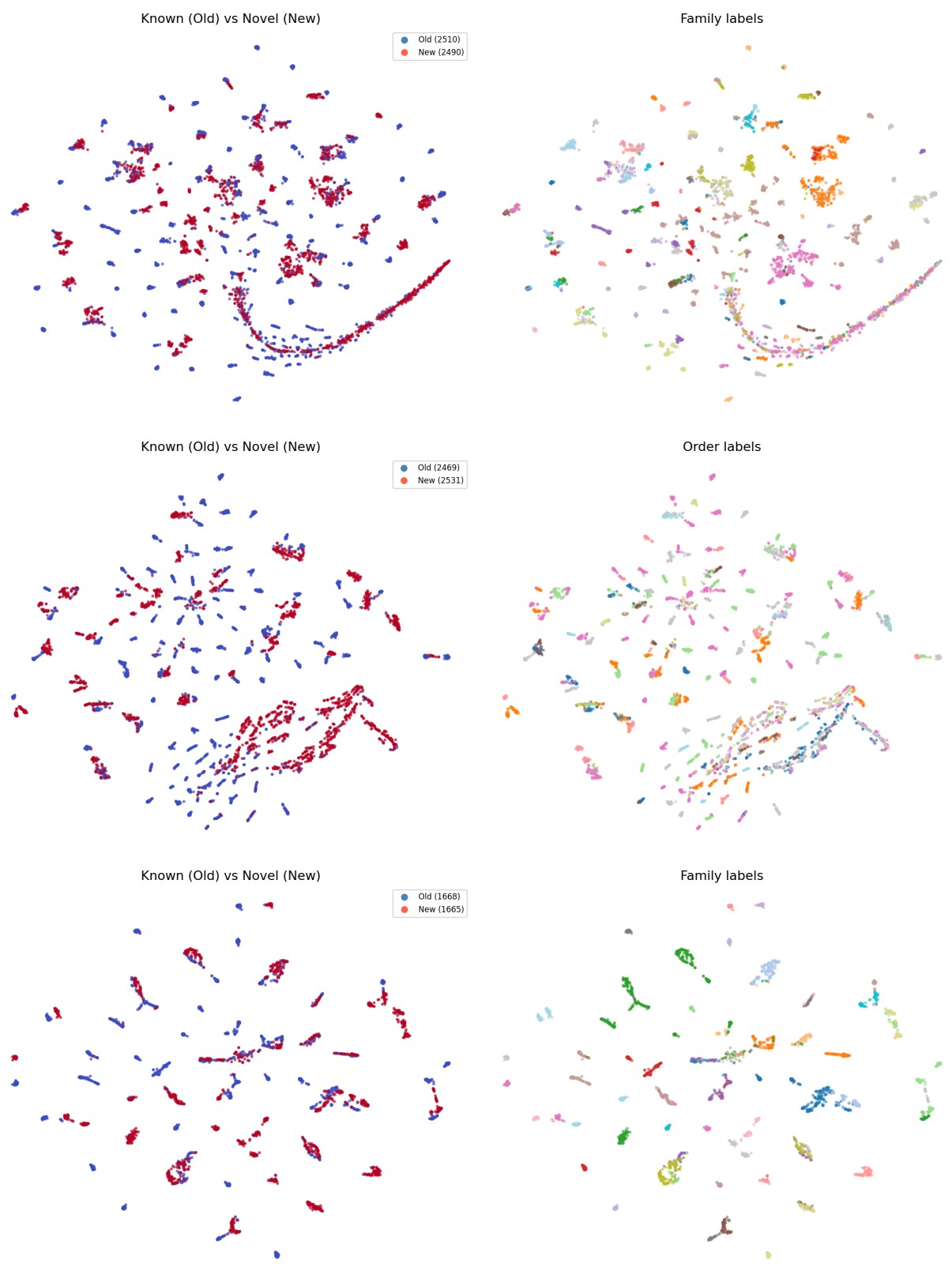

Figure 5: **t-SNE of the JS variant** on CUB (top), Stanford Cars (middle) and FGVC-Aircraft (bottom). JS yields cluster structure comparable to $W_1$ on CUB and Stanford Cars, and cleaner novel-class clusters on FGVC-Aircraft, consistent with its higher New accuracy there (Table 9).

