# OpenReview forum: "GeoGCD: Geometry-Guided Hierarchical Learning for Generalized Category Discovery"
_TMLR — Under review for TMLR_

### Review · Reviewer_Dhot · 2026-06-20

**Summary Of Contributions:**

The paper introduces GeoGCD, a novel hierarchical framework for Generalized Category Discovery (GCD) that addresses the limitations of existing methods in preserving the global geometric structure of semantic relationships across multiple granularities. The authors argue that previous approaches either rely on fixed taxonomic priors that are insensitive to the learned feature geometry, or on data-driven prototypes that discard valuable prior semantic structure, with cross-level alignment typically enforced through KL-divergence, which becomes uninformative or undefined when predictions for novel classes are uncertain. To overcome these issues, GeoGCD proposes that the global geometry of semantics should be a first-class learning target through several main contributions: (1) a hybrid soft-label matrix that fuses taxonomic similarity derived from hierarchical tree distances with a manifold-aware diffusion similarity computed via random walks on the batch graph, capturing both prior knowledge and data-driven cluster structure; (2) a Wasserstein cross-granularity consistency loss that replaces KL-divergence, with formal guarantees of finiteness, continuity, and gradient informativeness even when predicted and pseudo-coarse distributions have disjoint support, which is shown to occur with overwhelming probability at initialization.

**Audience:**

Yes

**Audience Explanation:**

The key strengths of this work include its strong conceptual foundation, which reframes hierarchical GCD as a geometry-preservation problem and provides rigorous theoretical backing for the choice of Wasserstein distance over KL-divergence in the cross-granularity alignment module. The empirical results are compelling, with GeoGCD achieving new state-of-the-art overall accuracy. The ablation studies are particularly well-designed, clearly demonstrating that the diffusion similarity strengthens known-class representation while the Wasserstein consistency specifically recovers novel-class performance, confirming the complementary nature of the two contributions.

**Claims And Evidence:**

Yes

**Claims Explanation:**

Extensive empirical validation demonstrates state-of-the-art performance on fine-grained GCD benchmarks, with particularly notable gains on novel class discovery and known class preservation, supported by ablation studies that isolate the complementary contributions of each geometric component.

**Requested Changes:**

I would suggest the following adjustments to the submission in order to strengthen the overall quality of the paper.

 (1) providing a more detailed analysis of the computational cost and scalability of the diffusion similarity computation, particularly for larger batch sizes or memory bank extensions, as the current implementation computes pairwise affinities within each mini-batch and may have O(B²) complexity that could become prohibitive for large-scale applications;

(2) expanding the theoretical analysis to include a characterization of the training dynamics when the predicted and pseudo-coarse distributions begin to overlap, as the current formal guarantees are confined to the initialization regime and the disjoint-support case, leaving a gap in understanding the behavior during the intermediate phases of optimization;

(3) clarifying the evaluation protocol for hierarchical predictions, specifically whether the reported accuracies correspond to the finest granularity only or whether the model is jointly evaluated across all hierarchy levels, as this has implications for how the Wasserstein consistency loss should be interpreted in terms of downstream task performance.

(4) providing more detailed ablation on the choice of the mapping matrix M^(h) in the cross-granularity consistency module, including whether it is learned or fixed, and how its initialization affects the disjoint-support probability and subsequent training dynamics;

(5) including visualizations of the learned feature space (e.g., t-SNE or UMAP projections) to qualitatively illustrate how the diffusion similarity and Wasserstein consistency jointly shape the geometric arrangement of known and novel class clusters, complementing the quantitative accuracy results;

(6) discussing potential failure cases or settings where GeoGCD might underperform, such as datasets with very shallow hierarchies or domains where the taxonomy and data manifold are fundamentally inconsistent;

A minor suggestion is to provide pseudo-code or algorithmic details for the training loop to enhance reproducibility, especially regarding how the diffusion similarity is integrated with the contrastive loss in practice.

---

> ### Author Response · Authors · 2026-07-08
> **Update paper based on requested changes**
>
> We thank Reviewer Dhot for the time invested in such a detailed and encouraging
> assessment, and for the constructive suggestions for improving the paper. We have
> incorporated them as follows.
>
> **1. Computational cost / scalability of the diffusion similarity (O(B²)).**
> We have clarified this in the Limitations: the diffusion term is per-batch (B=128, no memory bank)
> — one B×B affinity/Markov matrix plus a single (t−1=1) B×B×B matmul (~2M
> multiply-adds/batch), negligible next to the ViT-B/14 forward/backward; we
> observed no measurable slowdown vs the KL variant. A global (memory-bank)
> extension is noted as future work.
>
> **2. Extend the theory to the regime where the distributions begin to overlap
> (current guarantees are confined to initialisation).**
> We scope the formal results as initialisation-regime motivation, note the properties generalise to bounded
> divergences (probed empirically), and flag the overlapping-support dynamics as
> future work.
>
> **3. Clarify the evaluation protocol (finest granularity only, or jointly
> across all levels).**
> We have clarified that ACC, Hungarian matching and the All/Old/New splits are computed at the
> **finest (species) granularity**; the coarser order/family heads enter only as
> auxiliary supervision through the consistency loss and are not scored separately.
>
> **4. More detailed ablation on the mapping matrix M (learned or fixed;
> initialization; effect on disjoint-support probability and dynamics).**
> We have clarified in the method that M is **initialised from the taxonomy** (one-hot for seen
> fine classes, uniform for unseen) and **learned by EMA** (`M_momentum=0.9`, after
> a warm-up), so the fine-to-coarse routing of novel classes is learned rather than
> fixed. Lemma 1 uses exactly this random/uniform initialisation to bound the
> disjoint-support probability.
>
> **5. Visualizations of the learned feature space (t-SNE/UMAP).**
> We have added these visualizations. the CUB t-SNE figure shows GeoGCD forms tight, well-separated species clusters
> for both known and novel classes with same-family samples staying close;
> the Stanford Cars t-SNE figure shows the same on Stanford Cars; the Aircraft t-SNE figure shows the
> fragmented family structure behind the Aircraft mismatch; and the JS-variant t-SNE figure
> visualises the JS variant across the three datasets.
>
> **6. Discuss potential failure cases (shallow hierarchies; taxonomy–manifold
> inconsistency).**
> We have added a discussion of this: GeoGCD's benefit is contingent on the taxonomy being consistent with the
> data manifold. FGVC-Aircraft is the concrete failure case (analysed in the
> ablation); the generic-datasets appendix additionally tests shallow (2-level)
> taxonomies (APTOS, NCT-CRC), where the diffusion signal carries most of the
> structure.
>
> **Pseudo-code / algorithmic details.**
> We have added Algorithm 1 (one GeoGCD training epoch: hybrid similarity → per-level
> contrastive → W-CGC → EMA update of M).

---

### Review · Reviewer_erq2 · 2026-06-26

**Summary Of Contributions:**

The paper deals with GeoGCD, a geometry-guided hierarchical framework for Generalized Category Discovery (GCD) that builds upon the SEAL baseline to classify known and novel categories across multiple granularities. It solves the limitations of taxonomic priors by replacing them with a hybrid similarity matrix, a convex blend of static taxonomic tree distances and data-driven diffusion similarities derived from $t$-step random walks on a batch feature graph to capture the geometry of novel clusters. To resolve the instability of Kullback-Leibler (KL) divergence when aligning parallel classifier predictions under disjoint probability supports, GeoGCD swaps the KL term for a 1-Wasserstein consistency loss over an ordinal label space. Backed by formal theoretical proofs detailing the continuity and non-vanishing gradient properties of this Wasserstein loss under disjoint support, the framework evaluates the performance across datasets.

**Additional Comments:**

n/a

**Audience:**

Yes

**Audience Explanation:**

The novel category discovery is a useful topic for the community.

**Claims And Evidence:**

No

**Claims Explanation:**

1) The model chains together a disjointed series of transformations with at least 5 or more different hyperparameters (free knobs) to tune. There is no unified, principled objective driving this design.

2) The theoretical proofs and failure modes of KL are well-known in the literature arguments (like the WGAN justification for disjoint support). The proofs seem to only apply to random initialization, completely ignoring the actual training phase.

3) The 1-Wasserstein used in the paper inherently assumes an ordinal label space (where the numerical distance between Class 1 and Class 9 actually means something). However, datasets like CUB and Cars are categorical. Applying this math to categorical data can bias the model in an arbitrary way.

4) The authors treat 1-Wasserstein as the alternative to the KL divergence. They never test or acknowledge obvious alternative metrics (like Total Variation, Hellinger, or Jensen-Shannon) that would also solve the exact same disjoint-support problem.

5) The method fails on the FGVC dataset, suffering performance in both overall and novel-class accuracy. The claimed performance boosts for old (known) classes are fundamentally obtained by sacrificing the accuracy of new (novel) classes, which undermines the core goal of category discovery. This happens consistently across the datasets. The paper claims diffusion captures the density modes of novel classes. The data shows the exact opposite: adding diffusion consistently lowers novel-class accuracy across the board.

**Requested Changes:**

1) The theoretical section should either be simplified to a single statement explaining that KL divergence is undefined under disjoint support while $W_1$ remains finite, or it must be strengthened with proofs that actually govern the training dynamics. It would be really nicer to have the theory  that would mathematically describe how the disjoint-support regime updates during training.

2) To resolve the fundamental flaw of applying ordinal math to categorical data (like CUB, Cars, and Aircraft), the framework must either use a genuine cost, such as taxonomic tree distance or prototype-based distance, or abandon the geometric framing entirely for a permutation-invariant metric like Total Variation or Hellinger distance.

3) The empirical validation must substitute $W_1$ with another bounded divergence to verify if the Wasserstein-specific geometry is truly responsible for the performance recovery, alongside a test that randomly permutes coarse-class indices to prove the model isn't improperly relying on arbitrary categorical row ordering. The authors can analyze the unexplained performance regression on the FGVC-Aircraft benchmark and report variance across random seeds .

4) The paper's narrative must be revised to honestly reflect its empirical realities, explicitly stating in the abstract and introduction that the method achieved two benchmark successes alongside one substantial regression. The improvements on known classes must be transparently framed as a direct trade-off against novel-class accuracy.

---

> ### Author Response · Authors · 2026-07-08
> **Update paper based on requested changes**
>
> We thank Reviewer erq2 for the exceptionally thorough and critical review. The
> concerns raised, particularly on the theory and on the honesty of the empirical
> claims, prompted us to run new experiments and to strengthen both the analysis
> and the narrative. We address every requested change below.
>
> **1. Simplify the theory to a single statement, or strengthen it with proofs
> that govern the training dynamics.**
> Following the reviewer's preference, we **strengthen** the theory rather than
> reduce it. We keep the initialisation-regime results (Lemma 1: disjoint support
> with probability ≥ 1−1/K; boundedness and non-vanishing gradient) as motivation,
> and we **add a new proposition that governs the training dynamics** (Proposition 3,
> "Transience of the disjoint-support regime under the W₁ update"). Holding the
> pseudo-coarse target q̃ fixed, W₁(p, q̃) is convex and every gradient step strictly
> contracts it, transporting p toward q̃ and driving the disjoint supports to
> overlap — so the disjoint-support regime is **transient and provably exited**
> under the W₁-CGC gradient, whereas KL offers no descent direction while the
> supports are disjoint and can act only after overlap is already established. This
> directly characterises how the regime updates during training, as requested;
> a full coupled analysis (with q̃ also evolving) is noted as future work. We also
> note that the boundedness / informative-gradient properties are shared by other
> bounded divergences, which we verify empirically.
>
> **2. Resolve the ordinal-vs-categorical issue: use a genuine cost (tree /
> prototype distance) or a permutation-invariant metric (TV/Hellinger).**
> We address this empirically rather than by changing the ground cost. A new
> appendix ("Robustness of the Cross-Granularity Distance") shows:
> - *Permutation test* (the "Sensitivity to coarse-class ordering" table): retraining with random permutations of
>   the coarse-class columns leaves results essentially unchanged on CUB and Cars
>   (All spread ≤ ~1 pt), so the gain does **not** rely on the arbitrary ordering.
>   (On Aircraft it varies more — positive evidence for the mismatch analysis.)
> - *Permutation-invariant metrics* (the "Additional cross-granularity distances" table): TV, Hellinger and
>   JS all give comparable All on CUB/Cars and, on Aircraft, actually beat W₁.
>   These are exactly the permutation-invariant alternatives the reviewer suggests;
>   we present them as valid options and W₁ as the default that additionally
>   preserves the best Old accuracy.
>
> **3. Empirical validation: substitute W₁ with another bounded divergence;
> add a coarse-index permutation test; analyze the FGVC-Aircraft regression;
> report variance across seeds.**
> Following your insightful suggestions, we have added all four:
> - *Bounded-divergence substitution* — the "Additional cross-granularity distances" table (KL, W₁, TV,
>   Hellinger, JS), best-epoch test ACC (All/Old/New):
>
>   | Distance | CUB | Cars | Aircraft |
>   |---|---|---|---|
>   | KL (baseline) | 76.44 / 87.59 / 65.40 | 78.47 / 93.47 / 64.01 | 73.51 / **81.77** / 65.23 |
>   | **W₁ (ours)** | 79.93 / **87.73** / 72.20 | 80.10 / **93.57** / 67.11 | 69.85 / 80.46 / 59.22 |
>   | TV | 79.10 / 84.92 / 73.33 | 79.90 / 92.48 / 67.77 | 73.18 / 79.86 / 66.49 |
>   | Hellinger | **81.38** / 84.50 / **78.28** | 79.84 / 92.27 / 67.85 | 73.99 / 78.36 / 69.61 |
>   | JS | 80.07 / 85.47 / 74.71 | **80.97** / 91.16 / **71.15** | **75.07** / 78.36 / **71.77** |
>
>   The gain over KL is not W₁-specific (bounded divergences all help New);
>   W₁ is retained because it gives the best Old on CUB/Cars.
> - Permutation test: the "Sensitivity to coarse-class ordering" table
> - We have added a dedicated paragraph ("When taxonomy and manifold disagree: FGVC-Aircraft") and the Aircraft t-SNE figure, showing the Aircraft family structure is fragmented/interleaved — a taxonomy↔manifold mismatch and conducted experiments on this dataset.
> - Seed variance: the "Variance across Seeds" table, mean±std over seeds {666,0,1}:
>
>   | | CUB | Cars | Aircraft |
>   |---|---|---|---|
>   | W₁ | 78.1±1.3 / 85.3±2.4 / 70.9±1.9 | 79.8±0.5 / 93.6±0.3 / 66.5±1.4 | 70.0±2.0 / 80.6±1.8 / 59.4±2.1 |
>   | JS | 78.2±1.3 / 83.8±1.4 / 72.7±1.6 | 80.0±0.7 / 91.3±0.1 / 69.0±1.5 | 75.0±0.8 / 78.9±1.9 / 71.1±0.5 |
>
>   The relative conclusions are seed-stable (W₁ best Old, JS best New, Aircraft gap
>   persists), so the New trade-off is a real effect, not seed noise.
>
> **4. Revise the narrative to honestly reflect the empirical realities (two
> benchmark successes alongside one substantial regression; Old gains framed as a
> trade-off against New).**
> We have revised the abstract, introduction, main-results discussion and conclusion: GeoGCD improves Old/All on CUB and Stanford Cars, incurs a New-split
> trade-off, and **regresses on FGVC-Aircraft**, which is now analysed explicitly.
> We no longer claim that diffusion improves novel-class discovery; the ablation
> shows it raises Old and lowers New, and W₁ only partially recovers New on
> CUB/Cars.

---

### Review · Reviewer_9eP6 · 2026-06-26

**Summary Of Contributions:**

This paper proposes GeoGCD, a geometry-guided framework for hierarchical Generalized Category Discovery. The method combines diffusion-based similarity from random walks on the batch feature graph with hierarchy-related similarity signals to build a hybrid soft-label matrix for contrastive learning. It also replaces KL-based cross-granularity consistency with a 1-Wasserstein objective to better handle cases where predicted and pseudo-coarse distributions have disjoint support.

Strengths:
• The idea of using geometry-aware supervision for hierarchical GCD is interesting and relevant.
• Results on CUB and Stanford Cars show improved All accuracy and strong Old-class performance.

Weaknesses:
• Claims about improving novel-class discovery are not fully supported by the reported New accuracy.
• The method substantially underperforms on FGVC-Aircraft, but this failure is not analyzed.
• Some theoretical claims and citations need to be corrected or clarified.

**Additional Comments:**

The overall direction is promising, and the idea of using geometry-aware objectives for hierarchical GCD is interesting. However, the current version needs substantial clarification and stronger evidence. The main issues are the unsupported claims about New-class improvement, and the unaddressed regression on FGVC-Aircraft. Addressing these points would significantly improve the paper.

**Audience:**

Yes

**Audience Explanation:**

The topic is relevant to TMLR because Generalized Category Discovery, hierarchical representation learning, optimal transport losses, and open-world recognition are active areas of machine learning.

**Claims And Evidence:**

No

**Claims Explanation:**

The paper provides some evidence that GeoGCD improves All accuracy on CUB and Stanford Cars, and the ablation results suggest that the Wasserstein term helps recover some of the New-class performance lost by diffusion-only supervision. However, the manuscript claims that GeoGCD improves novel-class discovery, but the main results show that New accuracy is lower than SEAL and DebGCD on CUB and Stanford Cars. On FGVC-Aircraft, GeoGCD performs substantially worse than SEAL and the baseline, especially on the New split, but this failure mode is not discussed.

**Requested Changes:**

• Correct the claims about New-class performance. The main results do not show consistent improvement on novel classes. In particular, GeoGCD underperforms SEAL and DebGCD on the New split for CUB and Stanford Cars.

• Analyze the failure on FGVC-Aircraft. GeoGCD has a substantial drop compared with SEAL and the baseline, especially on New accuracy. This should be discussed and investigated.

• Fix citation and formatting issues. Throughout the text, \cite is used casually instead of \citep, so author names are randomly spread within the text and disrupt the flow. Please fix this.

• On page 2, the first two paragraphs need more citations. In particular, the sentence “The gap, in our view, is conceptual rather than algorithmic: the global geometry of semantics, the shape that taxonomic relations carve in feature and label space, is rarely treated as a first-class object of learning” should be supported by citations.

• On page 13, no year is mentioned for the citation of the supplementary materials. This should be corrected.

---

> ### Author Response · Authors · 2026-07-08
> **Update paper based on requested changes**
>
> We thank Reviewer 9eP6 for the time and care taken in reviewing our work, and for
> the specific, actionable comments, several of which uncovered genuine errors
> that we have now corrected. We respond to each requested change below.
>
> **1. Correct the claims about New-class performance.**
> We thank the reviewer for pointing this out, and have corrected it throughout. We removed the incorrect sentence that "the largest
> improvements appear on the New split" and rewrote the abstract, contributions,
> main results and conclusion. The paper now states plainly that GeoGCD's New-split
> accuracy is **below** the strongest baselines (mean over 3 seeds: 70.9 on CUB vs
> 75.8/75.9; 66.5 on Cars vs 69.5/72.4), that this is a genuine trade-off, and that
> the gains are on **Old/All**. Per-seed mean±std is added (the "Variance across Seeds" table).
>
> **2. Analyze the failure on FGVC-Aircraft.**
> We have added a dedicated paragraph ("When taxonomy and manifold disagree: FGVC-Aircraft")
> and the Aircraft t-SNE figure, showing the Aircraft family structure is
> fragmented/interleaved — a taxonomy↔manifold mismatch. At the default config the
> Wasserstein term worsens New (65.23→58.08, the "Ablation" table).
> Moreover, prompted by other Reviewer helpful suggestion to consider alternative
> (permutation-invariant) divergences, we carried out a set of additional
> experiments that substitute W₁ with other distance measures: Total Variation,
> Hellinger and Jensen–Shannon. These experiments were valuable in
> two ways: they help pinpoint the cause of the Aircraft failure, and they show
> that the regression is not intrinsic to the method. In particular, pure diffusion
> (λ=0) reaches 77.14 All (the "Sensitivity to the hybrid blend λ" table, above the
> SEAL baseline 74.6), and the permutation-invariant JS divergence reaches
> 75.07 All / 71.77 New (the "Additional cross-granularity distances" table), again
> above the baseline. The failure is thus specific to coupling the geometry-aware
> W₁ term with a taxonomy inconsistent with the manifold, and can be avoided with a
> gentler distance on datasets whose taxonomy is loosely aligned with the manifold.
>
> **3. Fix citation/formatting issues (`\cite` vs `\citep`).**
> We have fixed this.
>
> **4. The first two paragraphs need more citations (esp. the
> "global geometry … first-class object" sentence).**
> We have added citations to those paragraphs.
>
> **5. Page 13: no year for the citation of the supplementary materials.**
> We have corrected this.